# F*in*P: Fairness-in-Privacy in Federated Learning by Addressing Disparities in Privacy Risk

## Abstract

Ensuring fairness in machine learning extends to the critical dimension of privacy, particularly in human-centric federated learning (FL) settings where decentralized data necessitates an equitable distribution of privacy risk across clients. This paper introduces F*in*P, a novel framework specifically designed to address disparities in privacy risk by mitigating disproportionate vulnerability to source inference attacks (SIA). F*in*P employs a two-pronged strategy: (1) server-side adaptive aggregation, which dynamically adjusts client contributions to the global model to foster fairness, and (2) client-side regularization, which enhances the privacy robustness of individual clients. This comprehensive approach directly tackles both the symptoms and underlying causes of privacy unfairness in FL. Extensive evaluations on the Human Activity Recognition (HAR) and CIFAR-10 datasets demonstrate F*in*P's effectiveness, achieving improvement in fairness-in-privacy on HAR and CIFAR-10 with minimal impact on utility. F*in*P improved group fairness with respect to disparity in privacy risk using equal opportunity in CIFAR-10 by 57.14% compared to the state-of-the-art. Furthermore, F*in*P significantly mitigates SIA risks on CIFAR-10, underscoring its potential to establish fairness in privacy within FL systems without compromising utility.

## 1 Introduction

The increasing deployment of machine learning (ML) across diverse human-centric applications necessitates a rigorous examination of its fairness and ethical implications (Mehrabi et al., 2021). While significant attention has been devoted to algorithmic bias and ensuring equitable outcomes (Kleinberg et al., 2018; Rambachan et al., 2020; Dwork et al., 2012; Kusner et al., 2017; Zhao et al., 2024), a critical yet often overlooked dimension is the **fairness in the privacy** risks imposed on individuals participating in ML systems. A recent example is the 2024 National Public Data (NPD) breach, which exposed billions of records, underscoring the critical need for fairness in privacy. While affecting millions, the breach disproportionately impacted vulnerable populations like low-income individuals, the elderly, and those with disabilities, who are more susceptible to the consequences of data breaches (Staff, 2024). This highlights a key limitation of traditional privacy approaches that focus on average risk, neglecting equitable distribution.

This lack of equitable distribution is particularly salient in federated learning (FL), a paradigm designed for privacy-preserving collaborative learning on decentralized data (McMahan et al., 2017a). In FL, the inherent heterogeneity in client data distributions and system capabilities can lead to a disparate landscape of privacy vulnerabilities, where certain clients face disproportionately higher risks of privacy leakage than others (Shokri et al., 2017). A key threat exacerbating this privacy unfairness is the source inference attack (SIA) (Hu et al., 2021). By analyzing the shared global model, an adversary can infer whether a specific client contributed to its training. This capability poses a significant privacy risk and, crucially, can manifest unevenly across clients, leading to a violation of fairness in privacy. Clients with more unique data may be more easily identifiable, thus bearing a greater privacy burden.

**Contributions.** To address this critical challenge of privacy unfairness in FL, we introduce F*in*P: Fairness-in-Privacy, a novel framework designed to mitigate disparities in privacy risk by focusing on reducing the disproportionate vulnerability to SIA. Our approach is two-fold, tackling both the symptoms and the root causes of privacy unfairness: (1) a server-side adaptive aggregation strategy

that dynamically modulates client contributions to the global model to promote fairness in both the learning process and its resulting privacy implications, and (2) client-side regularization techniques designed to enhance the inherent privacy robustness of individual clients against inference attacks, mitigating individual client vulnerabilities. The efficacy of F*in*P is evaluated on two widely used datasets: Human Activity Recognition (HAR) and CIFAR-10. Our experimental results demonstrate that F*in*P achieves improvement in fairness in privacy, as measured by the Coefficient of Variation of SIA accuracy. These findings underscore F*in*P's potential to establish a more fair and privacy-preserving FL ecosystem without substantial performance degradation.

## 2 BACKGROUND AND RELATED WORK

**Privacy of Human-Centric Systems.** Ensuring privacy in human-centric ML-based systems presents inherent conflicts among service utility, cost, and personal and institutional privacy (Szti-panovits et al., 2019). Without appropriate incentives for societal information sharing, we may face decision-making policies that are either overly restrictive or that compromise private information, leading to adverse selection (Jin et al., 2016). Such compromises can result in privacy violations, exacerbating societal concerns regarding the impact of emerging technology trends in human-centric systems (Mulligan et al., 2016; Fox et al., 2021; Goldfarb & Tucker, 2012). Consequently, several studies have aimed to establish privacy guarantees that allow auditing and quantifying compromises to make these systems more acceptable (Jagielski et al., 2020; Raji et al., 2020).Various studies focused on privacy-preserving ML techniques targeting human-in-the-loop decision-making systems (Abadi et al., 2016; Cummings et al., 2019; Taherisadr et al., 2023; Taherisadr & Elmalaki, 2024). Recognizing that perfect privacy is often unattainable, this paper examines privacy from a fairness perspective by ensuring a fair distribution of harm when privacy risk occurs, addressing the technical challenges alongside the ethical imperatives of fair privacy protection.

**Privacy Risks in Federated Learning (FL).** FL (McMahan et al., 2017a) is an approach in ML that enables the collaborative training of models across multiple devices or institutions without requiring data to be centralized. This decentralized setup is particularly beneficial in fields where data-sharing restrictions are enforced by privacy regulations, such as healthcare and finance. FL introduces new privacy challenges. A key threat is the Membership Inference Attack (MIA), where an adversary aims to determine if a specific data record was part of a model's training set (Shokri et al., 2017; Hu et al., 2022). MIA effectiveness has been demonstrated across various machine learning models, including FL, with adversaries able to infer, for example, if a specific location profile contributed to an FL model (Gu et al., 2022; Zhu et al., 2024). The Source Inference Attack (SIA) extends MIA by identifying the specific client owning a training record, posing a significant risk (Zhu et al., 2021).

**Fairness in FL.** Fairness in FL is crucial due to the varied data distributions among clients, which can lead to biased outcomes (Ezzeldin et al., 2023). Achieving fairness involves balancing the global model's benefits across clients despite the decentralized nature of the data. Approaches include group fairness, ensuring performance distribution fairness, which focuses on fair accuracy distribution (Selialia et al., 2024). Our work addresses a related but distinct challenge: ensuring a fair distribution of privacy risks across clients in FL. Specifically, we aim to prevent scenarios where certain clients are disproportionately vulnerable to privacy leakage.

## 3 PROBLEM STATEMENT

FL systems can be vulnerable to privacy attack from an "honest-but-curious" server. While adhering to the FL protocol, the server may attempt to infer sensitive client information by analyzing aggregated model updates. This can reveal private data points, patterns, or client identities. A key privacy threat is the two-stage attack consisting of MIA followed by SIA:

- **MIA:** The server aims to determine if a specific data point $x$ was used to train the global model $\theta_g$. Formally, $\text{MIA}(\theta_g, x) = \text{P}(x \in D_{\theta_g})$, where $\text{P}(x \in D_{\theta_g})$ is the probability that $x$ belongs to the training data $D_{\theta_g}$.
- **SIA:** If the MIA suggests $x$ was part of the training data, the server then identifies the contributing client $i$. Formally, $\text{SIA}(\theta_i, x) = \text{P}(\text{Client}_i \mid x, \theta_i)$, where $\text{P}(\text{Client}_i \mid x, \theta_i)$ is the probability that client $i$ contributed $x$ to the model $\theta_i$.

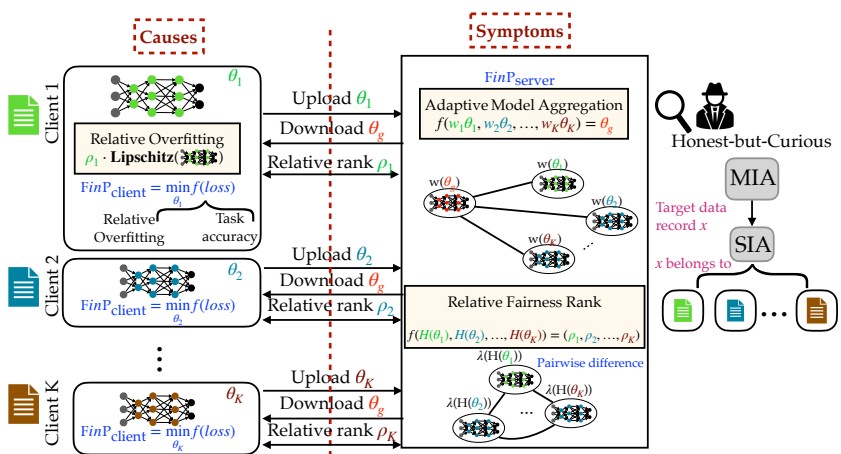

Figure 1: Fairness in Privacy F*in*P in FL by addressing the causes and the symptoms to achieve.

As demonstrated in prior work (Hu et al., 2021), the combination of MIA and SIA can severely compromise client privacy. Furthermore, auditing MIA has inherent limitations Chang et al. (2024). Our work addresses the disparity in privacy risk across clients. Given this threat model, where a compromised server can execute SIA attack, our objective is two-fold:

- **Addressing the Symptoms:** Develop a server-side aggregation method to ensure a fair distribution of privacy risk among clients.
- **Addressing the Causes:** Provide feedback to clients, enabling them to adjust their local updates to reduce overfitting and improve overall fairness in privacy.

We aim to mitigate the impact of SIA by enabling a fair distribution of the inherent privacy risk by introducing F*in*P framework. We assume an "honest-but-curious" server and cooperative clients capable of adjusting their local updates to enhance fairness in privacy .

## 4    FAIRNESS-IN-PRIVACY FRAMEWORK IN FEDERATED LEARNING

The core objective is that privacy risks should be distributed among all participating clients, preventing any single client from bearing a disproportionate burden. The heterogeneity of client data distributions, computational resources, and local training dynamics leads to disparities in privacy risks within federated learning systems. An overview of the F*in*P framework is shown in Figure 1. We argue that addressing fairness in privacy requires a two-pronged approach: handling it both at the server (during aggregation) and at the client (during local training). Server-side interventions, specifically adaptive aggregation, are crucial to mitigating the impacts of existing disparities in privacy risk. By carefully weighing client updates based on their estimated privacy risk, we can prevent highly vulnerable clients from unduly influencing the global model and further exacerbating the unfairness. However, server-side interventions alone are insufficient. They address the *symptoms* of unfairness but not the underlying *causes*.

The root cause of privacy disparity often lies in differences in local training dynamics, particularly local overfitting. Prior work has established the connection between overfitting and privacy risks in machine learning (Hu et al., 2021; Yeom et al., 2018; Shokri et al., 2017). Overfitting facilitates MIA and, under specific conditions, enables attribute inference attacks. Because overfitting can reveal information about the training data, it is a common vulnerability exploited by various inference attacks. Our framework F*in*P addresses fairness in privacy on the client side by introducing a **collaborative overfitting reduction strategy.** This strategy aims to proactively reduce the relative local overfitting, thereby minimizing the initial disparity in privacy risks before aggregation. By ranking clients based on their estimated relative overfitting and incorporating this rank into a local regularization scheme, we encourage clients to learn more generalizable representations, reducing their vulnerability to the disparity in privacy risk.

This two-pronged approach, combining adaptive aggregation at the server and collaborative overfitting reduction at the client, provides a comprehensive framework for achieving fairness in privacy in FL. By minimizing both the symptoms and the root causes of privacy disparity, our aim is to create a fairness in privacy within the FL system. This can be conceptually represented as a system with two primary interacting components. The server uses client models to calculate the adaptive weights for aggregation, and the clients use the feedback from the server to update their training. We formalize each component in the following Section 4.2 and Section 4.1 with detailed proof in Appendix A.

## 4.1 Formalizing Causes of Fairness in Privacy on Client Side

We formalize the fairness in privacy problem as follows: Given an FL system with $K$ clients and a global model $\theta_g$, our goal is to achieve fair privacy risk across all clients against successful SIA. To mitigate local overfitting (*causes*), we propose a collaborative client strategy. This leverages the fact that clients with higher overfitting are more susceptible to privacy risks.

The top Hessian eigenvalue ($\lambda_{\max}$) and Hessian trace ($H_T$) have been identified as important metrics for characterizing the loss landscape and generalization capabilities of neural networks (Jiang et al., 2020). Lower values of $\lambda_{\max}$ and $H_T$ typically indicate improved robustness to weight perturbations, leading to smoother training and better convergence. This is especially critical in FL, where the non-IID nature of data across clients creates distributional shifts that can exacerbate training instability and introduce fairness concerns. These distributional shifts can disproportionately impact certain client groups, leading to biased model performance (Mendieta et al., 2022). As we are interested in F*in*P, we determine each client's relative overfitting by calculating the average pairwise difference across the top Hessian eigenvalue ($\lambda_{\max}$) and Hessian trace ($H_T$):

$$\bar{\Delta}_k = \frac{1}{K-1} \sum_{j=1, j \neq k}^{K} |\lambda_{\max}^k - \lambda_{\max}^j|, \quad \bar{H}_k = \frac{1}{K-1} \sum_{j=1, j \neq k}^{K} |H_T^k - H_T^j|, \quad \rho_k = \frac{\frac{\bar{\Delta}_k}{\max \bar{\Delta}} + \frac{\bar{H}_k}{\max \bar{H}}}{2}. \tag{1}$$

The top Hessian eigenvalue of the local models of clients $k$ and $j$ are $\lambda_{\max}^k$ and $\lambda_{\max}^j$ respectively. Similarly, $H_T^k$, and $H_T^j$ are the Hessian trace of the local models of clients $k$ and $j$, respectively. We used the normalized average of both $\bar{\Delta}_k$ and $\bar{H}_k$ to quantify the client's *overfitting relative rank* ($\rho_k$), to serve as a proxy for relative privacy risk. Computing the Hessian eigenvalue and trace is done on the cloud server, and hence, there is no overhead of their computation on the client.

We incorporate this overfitting rank into the local training process using a regularization term based on the Lipschitz constant, approximated by the spectral norm of the Jacobian matrix ($||J_k||$) (Liu et al., 2020). In particular, a smaller Lipschitz constant implies smoother functions, less prone to overfitting, and better generalization. The modified local loss function for client $k$ is:

$$\mathcal{L}_k' = \mathcal{L}_k + \beta \cdot \rho_k \cdot ||J_k||, \tag{2}$$

$$\text{F}in\text{P}_{\text{client}} = \min_{\theta_k} \mathcal{L}_k' \tag{3}$$

where $\mathcal{L}_k$ is the original local loss function, $\rho_k$ is an adaptive controlling regularization strength that depends on the relative overfitting rank, $\theta_k$ are the local parameters of the client model that minimize the total loss $\mathcal{L}_k'$, and $\beta$ is the impact factor, which regularizes the impact of the Lipschitz constant on the learning task.

This penalizes models with large Lipschitz constants, promoting generalization and reducing overfitting. The regularization strength is weighted by $\rho_k$ **adaptively** at each round, applying stronger regularization to clients with higher overfitting ranks. This collaborative approach, using $\rho_k$ to guide local training, preserves privacy by reducing disparity in privacy risk. $\beta$ is a task-dependent parameter to regularize the loss $\mathcal{L}_k$ with Lipschitz loss. A larger $\beta$ improves fairness regularization but could make training unstable and fail to converge.

Computing the exact Hessian matrix is computationally challenging. Recent work in the literature explored multiple Hessian approximation techniques in FL to reduce the computational complexity, such as Empirical Fisher Information Matrix, Sketching Methods and Hessian-Vector Products. (Reddi et al., 2020; Ivkin et al., 2019) Using Hessian approximation may reduce the overfitting precision but avoid systemic bottleneck in computation complexity.

### 4.2 FORMALIZING SYMPTOMS OF FAIRNESS IN PRIVACY ON SERVER SIDE

We consider the privacy risk $p_k(\mathbf{w})$ for client $k$ to be influenced by the aggregation weights $\mathbf{w} = [w_1, w_2, ..., w_K]$, where $w_k$ represents the weight assigned for the client $k$, with the constraint $\sum_{k=1}^{K} w_k = 1$. This allows us to account for the varying client contributions to the global model. We define Fairness in Privacy (F$in$P) as minimizing the variance in privacy risks across clients. Our objective is to find the optimal weights for aggregation $\mathbf{w}$ that minimize the difference between individual client privacy risks and the average privacy risk. This is expressed in Equation 4.

$$\mathrm{F}in\mathrm{P}_{\mathrm{server}} = \min_{\mathbf{w} \in \mathcal{W}} \|\mathbf{p}(\mathbf{w}) - \frac{1}{K}\mathbb{1}^T \mathbf{p}(\mathbf{w}) \otimes \mathbb{1}\| + \|\frac{1}{K}\mathbb{1}^T \mathbf{p}(\mathbf{w})\|, \tag{4}$$

where $\mathbf{p}(\mathbf{w}) = [p_1(\mathbf{w}), \dots, p_K(\mathbf{w})]^T$ is the vector of privacy risks for all clients given the aggregation weights $\mathbf{w}$, $\mathbb{1}$ is a vector of ones of length $K$, and $\mathcal{W} = \{\mathbf{w} \in \mathbb{R}^K \mid \sum_{k=1}^{K} w_k = 1, w_k \geq 0 \, \forall k\}$ is the set of valid aggregation weights. The term $\frac{1}{K}\mathbb{1}^T \mathbf{p}(\mathbf{w})$ represents the average privacy risk. Equation equation 4 minimizes the Euclidean distance between individual privacy risks and this average, thus minimizing the disparity in privacy risks. Intuitively, we seek optimal aggregation weights to achieve a more equitable distribution of privacy risk, ensuring no client is disproportionately exposed.

We can quantify the *symptoms* of overfitting in the server by measuring the discrepancy between each client's local model update and the global model using the Principle Component Analysis (PCA) distance (Durmus et al., 2021). For client $k$, this distance, denoted as $p_k$, serves as a proxy for privacy risk; a larger $p_k$ signifies a symptom of greater overfitting and, thus, higher risk. Our proposed adaptive aggregation method aims to balance client contributions based on these PCA distances. By minimizing the variance of $p_k$ using the F$in$P$_{server}$ objective (Equation 4), we reduce the influence of clients exhibiting high overfitting (high $p_k$) and increase the influence of those with lower overfitting. This dynamic adjustment, performed in each FL round, enables a more fair distribution of privacy risk. We use PCA distances as one of the proxy for overfitting, which does not need any extra information except the client models. However, using PCA distance can be problematic because it's a fragile and unreliable proxy for privacy risk in high-dimensional neural networks such as ResNet. It computationally heavy in such spaces due to the "curse of dimensionality" and is sensitive to outliers (Candès et al., 2011). Hence, we also propose a lightweight FedAvg variant aggregation method called Adaptive Lightweight Aggregation(ALA) in server-side. Instead of using weighted sum of the local models based on dataset size of each client, we can weighted summarize local clients based on overfitting level $w_{global} \leftarrow \sum_{k=1}^{K} \left(\frac{1-\rho_k}{\sum_{j=1}^{K}(1-\rho_j)}\right) w^k$, where $\rho_k$ is the normalized overfitting level of client $k$. A higher $\rho_k$ close to 1 indicates worse overfitting. This lightweight aggregation method will not add any overhead in server side since there is no PCA calculation but need extra feedback on overfitting level $\rho_k$ from clients. The overfitting levels are calculated in client side and collected in server. We provide more details for the F$in$P framework in Appendix A.

## 5 EVALUATION

### 5.1 FEDERATED LEARNING SYSTEM SETUP

We evaluated F$in$P through two case studies, using the Human Activity Recognition (HAR) dataset (Section 5.4) and the CIFAR-10 image classification dataset (Section 5.5). For HAR, we compared four approaches: (1) a Baseline Federated Learning (FL) implementation using FedAvg, adapted from Hu et al. (2023); (2) F$in$P$_{server}$, which applies adaptive aggregation at the server without client collaboration (Equation 4); (3) F$in$P$_{client}$, which employs client-side collaboration to mitigate relative overfitting but omits adaptive server aggregation (Equation 2); and (4) the full F$in$P approach, incorporating both F$in$P$_{server}$ and F$in$P$_{client}$. In the CIFAR-10 evaluation, we compared three approaches: (1) the same Baseline FL using FedAvg from Hu et al. (2023); (2) FedAlign (Mendieta et al., 2022), a state-of-the-art FL method designed to address data heterogeneity in CIFAR-10; and (3) F$in$P approach using the same ResNet model as FedAlign (4) F$in$P approach using CNN with 4 different impact factor $\beta$=0.05, 0.1, 0.3, and 0.5. We evaluated F$in$P with the two different server-side aggregation strategies (PCA and ALA). More configurations details are placed in the Appendix B.

## 5.2 SIA Attack

We used the SIA setup explained in Hu et al. (2023). We randomly sampled training data from each client dataset and combined them in one dataset to be used as target records. This is a valid assumption, given an already successful Membership Inference Attacks (MIA) attack. In a practical scenario, an "honest-but-curious" server who knows the clients' identities and receives their model updates, could leverage this knowledge to trace training data back to its source, thus compromising client privacy. To launch SIA in FL setting, clients send their updated local model parameters to the server. The server uses each client's model to calculate the prediction loss on the target record. The client with the smallest loss is identified as the most probable source of that target record.

## 5.3 Metrics for Comparison

To comprehensively evaluate the effectiveness of F*in*P in achieving fairness in privacy, we employ the following key metrics encompassing privacy, fairness, utility, and efficiency.

**1- Fairness metrics:** We consider three metrics for fairness in privacy as follows:

• **Disparity in SIA accuracy:** We measure the dispersion of SIA accuracy across clients using the Coefficient of Variation (CoV) recognizing that fairness is related to the variance of shared utility rather than strict equality (Jain et al., 1984). A lower CoV indicates a more equitable distribution of privacy risk. For $K$ clients, we define the SIA accuracy for client $i$ as $\text{SIA}_{acc_i}$ calculated as:

$$\text{SIA}_{acc_i} = \frac{\text{Number of times SIA attack correctly identifies client } i}{\text{Total number of target records attributed for client } i} \quad (5)$$

The mean SIA accuracy ($\mu$) is calculated as $\mu = \frac{1}{K}\sum_{i=1}^{K}\text{SIA}_{acc_i}$ and $\sigma$ is the standard deviation of SIA accuracies. The CoV of SIA accuracy $\text{CoV}(\text{SIA}_{acc})$ is then computed as Equation 6:

$$\text{CoV}(\text{SIA}_{acc}) = \frac{\sigma}{\mu} = \frac{\sqrt{\frac{1}{K}\sum_{i=1}^{K}(\text{SIA}_{acc_i} - \mu)^2}}{\mu} \quad (6) \qquad \text{FI}(\text{SIA}_{acc}) = \frac{1}{1 + \text{CoV}(\text{SIA}_{acc})^2} \quad (7)$$

A lower CoV indicates a more fair distribution of SIA accuracy across clients, suggesting greater fairness in privacy. To facilitate interpretation as a fairness percentage between 0 and 1 (where 1 represents perfect fairness), we use the Fairness Index (FI($\text{SIA}_{acc}$)) transformation as Equation 7. A FI value of 1 indicates perfect fairness (all clients have the same SIA accuracy), while lower FI values indicate increasing disparities in SIA accuracy among clients.

• **Equal opportunity for group fairness:** Equation 5 can be interpreted as the true positive rate (TPR) of the SIA for client $i$. Hence, we can apply the group fairness metrics, specifically equal opportunity, by considering those clients as protected groups (Hardt et al., 2016). The equal opportunity criterion requires that the TPR of SIA identification be equal across these groups. This can be formulated using the equal opportunity difference (EOD) metric across $K$ clients as shown in Equation 8. A lower EOD indicates better group fairness.

$$EOD = \max_{i,j} \|\text{SIA}_{acc_i} - \text{SIA}_{acc_j}\| \ \forall \ i,j \in K \quad (8)$$

• **Disparity in target record prediction loss (SIA confidence):** A key aspect of our evaluation, beyond reducing SIA accuracy disparity, is addressing inter-client loss differences. SIA exploit the correlation between low client-side prediction loss on target record and successful source identification. Clients exhibiting lower prediction loss are, on average, more confidently identified by the adversary to be the source of the target record. F*in*P aims to mitigate this vulnerability by minimizing these inter-client prediction loss differences on target records, thereby reducing the the attack confidence. We quantify this effect using the Coefficient of Variation of client losses with target records, denoted as CoV(Loss), and the Fairness index, denoted as FI(Loss), similarly as defined in Equation 6 and Equation 7.

**2- Privacy metrics:** We asses the overall SIA success. In particular, F*in*P aims to reduce disparities in SIA success rates and prediction losses across clients. However, simply increasing the SIA accuracy of less vulnerable clients to match that of the most vulnerable ones would not represent true privacy improvement. To ensure privacy protection improvement, we evaluate the average SIA success rate across all clients and all communication rounds (Mean($\text{SIA}_{acc}$)) and the highest SIA

Table 1: Results of HAR dataset using the two approaches of server aggregation.

| | Accuracy (%) | | Privacy Metrics (%)) | | Fairness Metrics | | | Efficiency |
|---|---|---|---|---|---|---|---|---|
| | Training | Testing | Mean($\text{SIA}_{acc}$) ↓ | Max($\text{SIA}_{acc}$) ↓ | CoV($\text{SIA}_{acc}$)/FI($\text{SIA}_{acc}$) | CoV(loss)/FI(Loss) | EOD ↓ | Converge |
| Baseline | 74.10 | 76.97 | 19.34 | 22.40 | 0.94/0.54 | 0.244/0.938 | 0.55 | round 9 |
| F$in$P$_{\text{client}}$ | 73.39 | 75.86 | 18.87 | 23.30 | 0.97/0.53 | 0.237/0.941 | 0.52 | round 9 |
| F$in$P$_{\text{server}}$ (PCA) | **75.01** | **77.84** | **18.57** | **21.60** | 0.99/0.52 | 0.246/0.937 | 0.56 | round 11 |
| F$in$P (PCA) | 72.73 | 75.22 | 19.70 | 23.50 | **0.89/0.57** | 0.235/0.942 | **0.52** | round 10 |
| F$in$P$_{\text{server}}$ (ALA) | 71.57 | 73.82 | 19.55 | 26.10 | 0.93/0.55 | **0.220/0.948** | 0.55 | round 11 |
| F$in$P (ALA) | 72.83 | 76.09 | 19.29 | 22.00 | **0.89/0.57** | 0.235/0.942 | 0.53 | round 10 |

success rate observed across all clients and rounds (Max($\text{SIA}_{acc}$)). Lower values for both indicate stronger protection against SIA attacks on average.

**3- Utility and Efficiency metrics:** We evaluate the overall global server performance on the learning task after convergence. For classification tasks, we used the accuracy percentage of the testing data.

**4- Efficiency metrics:** We evaluate the impact of F$in$P on the number of FL communication rounds needed for convergence.

## 5.4 EVALUATING F$in$P ON HAR DATASET

The results of the evaluation metrics for HAR are summarized in Table 1 which show that applying F$in$P can improve fairness in privacy with respect to SIA compared to the Baseline (Hu et al., 2023).

**Impact on the disparity of SIA accuracy among clients** Our results demonstrate an improvement in fairness in privacy. F$in$P achieves a CoV($\text{SIA}_{acc}$) of 0.89 and a FI($\text{SIA}_{acc}$) of 0.57, compared to the Baseline's CoV($\text{SIA}_{acc}$) of 0.94 and FI($\text{SIA}_{acc}$) of 0.54. This represents a reduction of 5.32% in CoV($\text{SIA}_{acc}$) and a 5.56% improvement in FI($\text{SIA}_{acc}$) compared to baseline, indicating that F$in$P can reduce the disparity of SIA accuracy. More visual results are in Appendix C.1 (Figure 3).

**Impact on equal opportunity** F$in$P achieves better group fairness. F$in$P shows a smaller EOD of 0.52, showing a 5.45% reduction compared with the baseline.

**Impact on the disparity of SIA confidence among clients** Our results demonstrate improvement in fairness with respect to the SIA confidence in prediction among clients represented as CoV(Loss) and FI(Loss). F$in$P achieves a CoV(Loss) of 0.235 and FI(Loss) of 0.942. This improvement in FI(Loss) compared to the Baseline FI(Loss) of 0.938, indicating that F$in$P enhances the fairness of SIA confidence in prediction among clients. As the FI(Loss) is already high, the small improvement is still valuable. More visual results can be found in Appendix C.1 (Figure 4).

**Impact on SIA overall success** F$in$P shows a marginal increase of less than 1.1% in Mean($\text{SIA}_{acc}$) and less than 0.4% in Max($\text{SIA}_{acc}$), these gains are secondary to the primary objective of fairness improvement. The key achievement of F$in$P is the demonstrably more fair distribution of privacy protection as observed in FI($\text{SIA}_{acc}$).

**Impact on utility and efficiency** F$in$P maintains competitive classification accuracy. The global model's testing accuracy only decreases by a 1.75%. A visual representation of the convergence of the model is shown in Appendix C.2(Figure 5) which shows that F$in$P takes $\approx 10$ rounds to converge compared to $\approx 9$ rounds in the Baseline.

**Contributions of server-side and client-side components to fairness** Isolating the server-side adaptive aggregation F$in$P$_{\text{server}}$(with PCA) revealed a nuanced impact on fairness metrics. While F$in$P$_{\text{server}}$ reduced the variation in PCA distance (PCA$_d$) by 1.3%, it also reduce the Mean($\text{SIA}_{acc}$) and Max($\text{SIA}_{acc}$) by 0.77% and 0.6%, respectively. However, it also resulted in a slight shift in both FI($\text{SIA}_{acc}$) and FI(Loss) by $-0.02$ and $-0.001$, respectively. This suggests that server-side adaptation alone (F$in$P$_{\text{server}}$) primarily influences the distribution of model distances and has a less direct impact on the fairness metrics themselves. This observation motivated the investigation of client-side factors, specifically the disparity in the overfitting among clients, to further enhance fairness. A visual representation of the variation in PCA values can be found in Appendix C.4Figure 6. Both of the PCA-based aggregation and adaptive lightweight aggregation (ALA) strategies demonstrate the similar contributions on improving the fairness in privacy. However the timing per round and computational overhead are different. By utilizing the lightweight aggregation in server side can avoid the major bottleneck on PCA computation. Specifically, ALA takes 17% of the time needed

Table 2: Results of CIFAR dataset using ResNet as the local model.

| | Accuracy (%) | | Privacy Metrics (%)) | | Fairness Metrics | | | Efficiency |
|---|---|---|---|---|---|---|---|---|
| | Training | Testing | Mean(SIA$_{acc}$) ↓ | Max(SIA$_{acc}$) ↓ | CoV(SIA$_{acc}$)/FI(SIA$_{acc}$) | CoV(loss)/FI(Loss) | EOD ↓ | Converge |
| Baseline | 78.39 | 76.45 | 30.86 | 38.52 | **0.33/0.90** | 0.67/0.69 | 0.28 | round 10 |
| FedAlign | 70.79 | 71.87 | 30.72 | 38.46 | 0.34/0.89 | 0.86/0.58 | 0.28 | round 14 |
| F$in$P(with PCA) | **80.23** | **78.47** | **10.07** | **10.67** | 0.35/0.89 | **0.44/0.83** | **0.12** | round 12 |

in PCA server aggregation. It indicates that ALA can achieve comparable improvement but with significant less time. More details on timing can be found in Appendix C.3

Combining F$in$P$_{server}$(with PCA) or F$in$P$_{server}$(with ALA) with F$in$P$_{client}$ resulted in even higher fairness gains in FI(SIA) and FI(Loss) compared to using F$in$P$_{client}$ alone. This indicates that F$in$P$_{server}$ contributes synergistically to the fairness improvements achieved by F$in$P$_{client}$ when both are employed. This improvement accounts for the adaptation of the aggregated weights **w** at the server-side (Equation 4) and the adaptive relative overfitting rank $\rho_k$, which is used as a regularizer strength at the client-side (Equation 2). More visual representations showing how PCA distance and relative overfitting rank $\rho_k$ change in every round in the FL are shown in Appendix C.4(Figure 7).

Analysis of Hessian eigenvalues ($\lambda_{max}$) and trace ($H_T$) revealed a strong correlation (Spearman's rank correlation coefficient $\approx 1$) between these two metrics, both indicative of how well a local model fits its local data. Based on this correlation, these metrics were given equal weight in Equation 1.

## 5.5 EVALUATING F$in$P ON CIFAR-10 DATASET

We evaluated F$in$P(with PCA) on CIFAR-10 using four different setups as described in Section 5.1. As observed in Table 2 F$in$P demonstrates a significant improvement in fairness in privacy, with competitive accuracy. The FI(Loss) for FedAvg (Hu et al., 2023), FedAlign (Mendieta et al., 2022), and F$in$P are 0.69, 0.58, and 0.83, respectively. In particular, F$in$P achieves a substantial increase in FI(Loss) of 20.3% compared to FedAvg and 43.1% compared to FedAlign. Notably, despite employing a distillation technique, FedAlign failed to effectively mitigate SIA risks, exhibiting a higher CoV(Loss) of 0.86 compared to FedAvg's 0.67. These results are further illustrated in Appendix D.1 Figure 11 and Figure 12.

While FedAlign and FedAvg exhibit similar Mean(SIA$_{acc}$) and Max(SIA$_{acc}$), F$in$P effectively mitigates these risks. As shown in Table 2, F$in$P reduces the Mean(SIA$_{acc}$) to 10.07%, approaching the random-guess probability of 1/10 (10%) for a 10-class classification task. Specifically, F$in$P reduces the Mean(SIA$_{acc}$) success rate from 30.86% to 10.07% and the Max(SIA$_{acc}$) from 38.52% to 10.67%. F$in$P demonstrates comparable CoV(SIA$_{acc}$) and FI(SIA$_{acc}$), yet shows a reduction in EOD from 0.28 to 0.12 compared with FedAvg and FedAlign. This improvement can be illustrated in Figure 9 in Appendix D which represent the average SIA accuracy across all clients in every round. Furthermore, Figure 10 in Appendix D that shows the reduction of EOD by approximately 57.14%.

Moreover, F$in$P maintains and slightly improves the classification accuracy. F$in$P achieves a testing accuracy of 78.46%, marginally higher than FedAvg's 77.62% with two extra FL rounds. More visual results can be found in Appendix D.2 Figure 13. This 0.84% improvement is attributed to the global model's aggregation of generalized client models through Lipschitz regularization rather than models overfit to individual datasets.

Additionally, We conducted ablation experiments on the impact factor $\beta$ to analyze its effect on model convergence and fairness metrics as summarized in Table 3. We evaluated both PCA and ALA aggregation strategies on server side and use CNN in client model. We observed that a larger $\beta$ can also improve testing accuracy due to improved generalization. However, excessive $\beta$ can be harmful to the model convergence as the Lipschitz constant may dominate the client training loss as demonstrated in the experiment with $\beta$=0.5, which makes the model fail to converge. More detailed data can be found in Appendix D.3. With lightweight aggregation in server (ALA), we observe it achieves similar fairness improvement and better testing accuracy, which demonstrate the ability of improving fairness in privacy while reducing computational complexity. This indeed comes as a tradeoff of sharing more information to the server as explained in Section 4.2. More results can be found in Appendix D.4.

Table 3: Ablation experiments on $\beta$. Results of CIFAR dataset using CNN as the local model (Hu et al., 2023).

| | Accuracy (%) | | Privacy Metrics (%)) | | Fairness Metrics | | | Efficiency |
|---|---|---|---|---|---|---|---|---|
| | Training | Testing | Mean(SIA$_{acc}$)↓ | Max(SIA$_{acc}$)↓ | CoV(SIA$_{acc}$)/FI(SIA$_{acc}$) | CoV(loss)/FI(Loss) | EOD↓ | Converge |
| Baseline Hu et al. (2023) | 75.62 | 62.37 | 40.91 | 46.70 | 0.23/0.95 | 0.46/0.83 | 0.29 | round 5 |
| F$in$P($\beta$=0.05, PCA) | 69.31 | 59.67 | 39.51 | 43.40 | 0.21/0.96 | 0.46/0.83 | 0.27 | round 5 |
| F$in$P($\beta$=0.1, PCA) | 70.45 | 61.19 | 39.47 | 43.70 | **0.19/0.96** | 0.44/0.84 | **0.25** | round 7 |
| F$in$P($\beta$=0.3, PCA) | 71.17 | 63.81 | **31.85** | **39.90** | 0.31/0.90 | **0.38/0.87** | 0.31 | round 12 |
| F$in$P($\beta$=0.5, PCA) | 10 | 10 | N/A | N/A | N/A | N/A | N/A | N/A |
| F$in$P($\beta$=0.05, ALA) | **76.03** | **64.26** | 38.62 | 43.90 | 0.23/0.95 | 0.46/0.83 | 0.29 | round 6 |
| F$in$P($\beta$=0.1, ALA) | 74.64 | 63.94 | 37.39 | 42.50 | 0.25/0.94 | 0.44/0.84 | 0.32 | round 7 |
| F$in$P($\beta$=0.3, ALA) | 71.00 | 64.09 | 34.09 | 41.00 | 0.34/0.89 | 0.41/0.86 | 0.38 | round 11 |

**Ablation experiments on local epochs and overfitting** Prior work (Hu et al., 2023; Yeom et al., 2018) indicates that in Federated Learning, greater data heterogeneity and more local training epochs increase model overfitting and the associated privacy risks. We conduct ablation experiments on 3 different local training epochs, 1, 5, and 10. Our results show that Mean(SIA$_{acc}$) and Max(SIA$_{acc}$) increase from 24.31%, 27.20% to 42.9% and 51.4% with local epochs 1 to 10. This indicates worse overfitting can result in a higher SIA risk. Visual results can be found in Appendix D.5 Figure 24.

## 5.6 Summary of F$in$P in HAR and CIFAR-10 results

In summary, our evaluation across HAR and CIFAR-10 datasets demonstrates the effectiveness of F$in$P in achieving fairness in privacy with respect to the disparity in the impact of source inference attacks (SIA) while maintaining model performance. Although F$in$P yields only a modest improvement on the HAR dataset, this is mainly due to the already high FI(loss) in this setting. Notably, on the CIFAR-10 dataset, F$in$P with ResNet effectively mitigates SIA risks approaching random-guess probability with 57.14% improvement in equal opportunity metric as detailed in Section 5.5. This result is attributed to the non-IID sampling of CIFAR-10 subsets across clients, where all data in each subset are part of the comprehensive CIFAR-10 dataset, and the adoption of ResNet. F$in$P promotes client model generalization and reduce disparity in prediction loss on target records, thereby neutralizing the effectiveness of source inference attacks (SIA).

## 6 Discussion and Limitations

**Differential Privacy against SIA** Prior works, including Hu et al. (2023), investigated the use of differential privacy (DP) (Dwork et al., 2006) as a defense mechanism against SIA in FL. Differential privacy was chosen due to its provable guarantees for privacy protection against inference attacks. However, their findings revealed that DP-SGD(Abadi et al., 2016) can severely degrade model utility with only a little decrease in SIA accuracy. Furthermore, Advanced client-level DP variants that better balance this trade-off typically require thousands of clients to be effective which may not be feasible in some setups(McMahan et al., 2017b). In contrast, F$in$P demonstrates that it is possible to effectively reduce the SIA success rate by addressing overfitting while maintaining model utility.

**Limitations** Our current evaluation relies on a specific type of privacy attack (SIA) and is evaluated over two datasets (HAR and CIFAR-10). Future work should investigate the effectiveness of F$in$P against other datasets and privacy attacks, such as attribute inference and model inversion. Furthermore, our client-side approach assumes a degree of client cooperation. Investigating mechanisms that incentivize or enforce client participation in the collaborative overfitting reduction strategy is an important direction for future research.

## 7 Conclusion

We proposed F$in$P framework which addresses a critical gap in federated learning (FL): the unequal distribution of privacy risks. Traditional FL prioritizes average privacy but often ignores disparities caused by data heterogeneity, resource differences, and local training dynamics. F$in$P tackles this through server-side adaptive aggregation and client-side collaborative overfitting reduction, promoting fairness-in-privacy. Our approach targets both the symptoms and causes of privacy inequality. Experiments showed a 57.14% improvement in fairness on the CIFAR-10 dataset and reduced SIA success rates to near random guess levels, with comparable testing accuracy.

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

## A  FORMAL PROOF OF FinP FRAMEWORK

**Proposition 1.** *A model with a smaller Lipschitz constant is less prone to overfitting.*

*Proof.* Let a neural network be a function $f_w : \mathcal{X} \to \mathcal{Y}$ parameterized by weights $w$. The Lipschitz constant $L$ of a function $f$ is the smallest number such that for all $x_1, x_2$ in the domain of f, we have:
$$\|f(x_1) - f(x_2)\|_2 \le L \|x_1 - x_2\|_2$$
The spectral norm of the Jacobian matrix, $\|J_w(x)\|_2$, of the network with respect to its input $x$ is equal to the local Lipschitz constant at that point. The global Lipschitz constant $L$ is bounded by the maximum spectral norm over the input domain: $L = \sup_{x \in \mathcal{X}} \|J_w(x)\|_2$.

An overfit model has memorized specific, often noisy, patterns in the training data. This implies that a small perturbation in the input can lead to a large change in the output, which is characteristic of a large Lipschitz constant. Conversely, a model with a small Lipschitz constant is inherently stable and less sensitive to minor input variations. It cannot "memorize" the idiosyncrasies of the training data because it is mathematically constrained to not react strongly to small perturbations. This forces the model to learn more generalizable features that are effective on unseen data.

By training with a regularization term that penalizes a large spectral norm of the Jacobian (as in FinP$_{client}$), we directly minimize the model's Lipschitz constant. This in turn reduces the model's sensitivity to input variations, making it less likely to overfit. □

**Proposition 2.** *A model that is overfit to a client's local data has a higher SIA accuracy.*

*Proof.* An SIA attacker identifies the source of a data record $x_t$ by finding the client $k$ whose model produces the minimum prediction loss on $x_t$. The SIA is successful if the identified client is the true source of $x_t$. The attack criterion is:
$$k^* = \arg \min_{k \in \mathcal{K}} L(f_k, x_t)$$
where $L(f_k, x_t)$ is the prediction loss of model $f_k$ on the target record $x_t$.

An overfit model $f_k^{\text{overfit}}$, by definition, has minimized the loss on its training data $\mathcal{D}_k$ to a great extent. For any data record $x_t \in \mathcal{D}_k$, the overfit model has effectively memorized it, resulting in a prediction loss close to zero.
$$L(f_k^{\text{overfit}}, x_t) \approx 0$$
For any other model $f_j$ where $j \ne k$, the data record $x_t$ was not part of its training set. Therefore, the loss of model $f_j$ on $x_t$ will be higher.
$$L(f_j, x_t) \gg L(f_k^{\text{overfit}}, x_t) \quad \forall j \ne k$$
Since the overfit model $L(f_k^{\text{overfit}}$ produces a uniquely low loss on its own data record $x_t$, it will almost always satisfy the SIA minimization criterion. This means the probability of a successful SIA against the overfit client is high. □

**Proposition 3.** *The FinP framework achieves fairness in privacy by minimizing the variance of privacy risks across clients. This is accomplished through a two-pronged approach that addresses both the root causes (FinP$_{client}$) and the symptoms (FinP$_{server}$) of privacy disparity.*

*Proof.* The proof for this proposition synthesizes the effects of both the FinP$_{client}$ and FinP$_{server}$components. We define the privacy risk of client $k$, denoted $p_k$, as its vulnerability to SIA, which is directly proportional to its SIA accuracy. Our objective is to prove that the combined FinP framework reduces the variance of the privacy risk vector $\mathbf{p} = [p_1, \ldots, p_K]$, thereby achieving fairness.

**Step 1: The Effect of FinP$_{client}$ (Addressing the Causes)** FinP$_{client}$ component targets the root cause of privacy disparity: overfitting. An overfit model is significantly more vulnerable to SIA because it produces a uniquely low loss on its own data, making it the clear choice for the attacker. The FinP$_{client}$ regularization term is defined as:
$$\mathcal{L}'_k = \mathcal{L}_k + \beta \cdot \rho_k \cdot \|J_k\|_2 \tag{9}$$
This regularization is adaptive due to the client's relative overfitting rank, $\rho_k$.

**Claim 1** (Adaptive Overfitting Reduction). *The FinP$_{client}$ component non-uniformly reduces overfitting, with a greater effect on clients who are most overfit, thereby reducing their individual privacy risks and compressing the upper tail of the privacy risk distribution.*

*Proof.* Let $p_k^{\text{baseline}}$ be the privacy risk of client $k$ in a standard FL setting. From prior proofs, we established that a model with a larger Lipschitz constant ($L$) is more prone to overfitting and, consequently, has a higher SIA accuracy. The spectral norm of the Jacobian, $\|J_k\|_2$, serves as a proxy for the Lipschitz constant, $L_k$. FinP$_{\text{client}}$ minimizes the loss function in Equation equation 9. The term $\rho_k \cdot \|J_k\|_2$ acts as a penalty on models with high Lipschitz constants. The value of $\rho_k$ is a relative overfitting rank.

- **For clients with high baseline overfitting**: $\rho_k$ is high. The product $\beta \cdot \rho_k$ is large, imposing a strong penalty that forces the optimizer to minimize the Lipschitz constant $L_k$. This directly reduces overfitting and, by extension, the client's privacy risk.

- **For clients with low baseline overfitting**: $\rho_k$ is small. The regularization term is negligible, and the optimization primarily focuses on the original loss $\mathcal{L}_k$. Their privacy risk remains approximately the same.

This targeted reduction pulls the most vulnerable clients closer to the average privacy risk, effectively compressing the distribution and reducing its variance. □

**Step 2: The Effect of FinP$_{\text{server}}$ (Addressing the Symptoms)** The FinP$_{\text{server}}$ component addresses the symptoms of privacy disparity by optimizing the aggregation weights $w$ to ensure a more fair distribution of risk at the server level. The objective function is:

$$\min_{\mathbf{w} \in \mathcal{W}} \|\mathbf{p}(\mathbf{w}) - \frac{1}{K}\mathbb{1}^T \mathbf{p}(\mathbf{w}) \otimes \mathbb{1}\|_2 + \|\frac{1}{K}\mathbb{1}^T \mathbf{p}(\mathbf{w})\|_2 \tag{10}$$

**Claim 2** (Variance Minimization via Aggregation). *The FinP$_{server}$ objective function explicitly minimizes the variance of the privacy risk vector $\mathbf{p}(w)$ by adjusting the aggregation weights $w$.*

*Proof.* Let $\mu_{\mathbf{p}(\mathbf{w})} = \frac{1}{K}\mathbb{1}^T \mathbf{p}(\mathbf{w})$ be the mean of the privacy risk vector. The objective function in Equation equation 10 is:

$$\min_{\mathbf{w} \in \mathcal{W}} \|\mathbf{p}(\mathbf{w}) - \mu_{\mathbf{p}(\mathbf{w})}\mathbb{1}\|_2 + \|\mu_{\mathbf{p}(\mathbf{w})}\mathbb{1}\|_2$$

The first term, $\|\mathbf{p}(\mathbf{w}) - \mu_{\mathbf{p}(\mathbf{w})}\mathbb{1}\|_2$, is the Euclidean distance from each client's privacy risk to the mean risk. Minimizing this term is equivalent to minimizing the sum of squared deviations from the mean:

$$\min_{\mathbf{w} \in \mathcal{W}} \sum_{k=1}^{K} \left( p_k(\mathbf{w}) - \mu_{\mathbf{p}(\mathbf{w})} \right)^2$$

Since variance is defined as $Var(\mathbf{p}(\mathbf{w})) = \frac{1}{K}\sum_{k=1}^{K}(p_k(\mathbf{w}) - \mu_{\mathbf{p}(\mathbf{w})})^2$, minimizing this sum is equivalent to minimizing the variance. The second term, $\mu_{\mathbf{p}(\mathbf{w})})^2$, minimizes the overall average privacy risk. Thus, the FinP$_{\text{server}}$ objective function explicitly seeks to minimize the variance of the privacy risk vector by assigning optimal aggregation weights $\mathbf{w}$. □

**Claim 3** (PCA Distance as a Proxy for Privacy Risk). *The PCA distance of a client's model update serves as a proxy for its SIA vulnerability.*

*Proof.* Let $\Delta \mathbf{w}_k$ be the model update vector from client $k$. The SIA vulnerability of a client is proportional to how "unique" its model update is, as a highly unique update is more likely to encode memorized, client-specific features. We can measure this uniqueness by how much the update deviates from the collective behavior of other clients.

PCA identifies the principal components that capture the maximum variance in a set of vectors. We can compute the principal components of the set of all client updates $\{\Delta \mathbf{w}_j\}_{j=1}^K$. The PCA distance

of a client's update $\Delta \mathbf{w}_k$ is the Euclidean distance from this vector to the subspace spanned by the top principal components.

A large PCA distance for client $k$ indicates that its model update $\Delta \mathbf{w}_k$ is an outlier and does not align with the common update directions of the other clients. The underlying claim that a large PCA distance indicates an outlier is mathematically sound. This divergence suggests that the model update is not contributing to the generalizable knowledge shared by the collective, but is instead focused on idiosyncrasies of its local data. Based on Proposition 2, this overfitting to local data directly corresponds to a high SIA vulnerability. Therefore, the PCA distance provides a computationally tractable way to quantify a client's privacy risk without having to perform a full-blown SIA.

However, a major drawback of using PCA distance as a privacy risk metric is its computational expense. Computing the principal components and the distance to the subspace requires singular value decomposition (SVD) of the matrix of client updates. The computational complexity of SVD is high, typically on the order of $\mathcal{O}(d^2 K)$, where $d$ is the dimension of the model parameters (which can be millions) and $K$ is the number of clients. In a large-scale federated learning setting with many clients and large models, this can be prohibitively slow and may not be feasible for real-time risk assessment at every communication round. This computational bottleneck is a key motivation for the development of the more lightweight adaptive FedAvg method as an alternative. $\qquad \square$

**Claim 4** (Adaptive Lightweight FedAvg as a Proxy for Variance Minimization). *The lightweight FedAvg aggregation method, by weighting client models based on their inverse overfitting rank, serves as a computationally efficient proxy for the FinP$_{server}$ variance minimization objective.*

*Proof.* Let the lightweight FedAvg aggregation weights be defined as:

$$w_k^{\text{light}} = \frac{1 - \rho_k}{\sum_{j=1}^{K}(1 - \rho_j)}$$

where $\rho_k \in [0, 1]$ is the normalized overfitting rank of client $k$. A higher $\rho_k$ indicates worse overfitting and a higher privacy risk.

We need to show that this method effectively minimizes the variance of the privacy risk vector $\mathbf{p} = [p_1, \ldots, p_K]$. From Proposition 2, we have established a direct, monotonic relationship between overfitting ($\rho_k$) and privacy risk ($p_k$), i.e., $p_k = f(\rho_k)$ where $f$ is a monotonically increasing function. Therefore, minimizing the variance of $p$ is equivalent to minimizing the variance of the vector of overfitting ranks, $\rho = [\rho_1, \ldots, \rho_K]$.

The lightweight aggregation weights $w_k^{\text{light}}$ are inversely proportional to the overfitting rank $\rho_k$. This means that for clients with high overfitting ($\rho_k \to 1$), their contribution to the global model update is significantly reduced ($w_k^{\text{light}} \to 0$). Conversely, for clients with low overfitting ($\rho_k \to 0$), their contribution is maximized.

Let's consider the impact of this aggregation on the overall model. The global model update is a weighted sum:

$$\Delta w_g = \sum_{k=1}^{K} w_k^{\text{light}} \Delta w_k$$

By reducing the influence of clients with high overfitting ranks, the aggregated model update is less influenced by the most overfit and privacy-vulnerable clients. This in turn reduces the overall privacy risk contributed by these clients to the global model, as their unique data characteristics are not significantly embedded in the final model.

While this method does not explicitly solve the optimization problem in Equation equation 10, it serves as a heuristic. It directly penalizes the most vulnerable clients by reducing their impact, thereby effectively compressing the distribution of privacy risks. The result is a lower variance in the privacy risk vector, analogous to the effect achieved by the more computationally intensive PCA-based method. $\qquad \square$

**Step 3: Synthesis of the Two Components** The true strength of the FinP framework lies in the synergistic interaction between the client and server components, creating a collaborative feedback loop in each FL round:

1. **Server Measures Disparity**: The server computes the relative overfitting ranks ($\rho_k$).

2. **Server Provides Feedback**: The server sends the computed $\rho_k$ values to the clients.

3. Client Reduces Risk: The clients use the received $\rho_k$ to apply targeted regularization, proactively reducing their own privacy risk. This action reduces the initial variance of the privacy risk distribution.

4. Server Balances Contributions: The server's adaptive aggregation, by minimizing the variance of the privacy risk vector (via either the explicit optimization or the lightweight FedAvg proxy), ensures that any remaining disparities in privacy risk are mitigated by assigning optimal aggregation weights.

$\square$

The F*in*P framework achieves Fairness in Privacy by combining a client-side mechanism that elevates the privacy of the most vulnerable clients with a server-side mechanism that limits their influence on the global model. This collaboration leads to a provable reduction in the variance of privacy risks across all clients.

$$\text{Var}(\text{Privacy Risk}_{\text{F}in\text{P}}) < \text{Var}(\text{Privacy Risk}_{\text{Baseline}})$$

## B    DATASETS AND SETUP

We evaluate F*in*P using the UCI HAR(Reyes-Ortiz et al., 2013) and CIFAR-10 datasets (Krizhevsky et al., 2009) utilizing federated learning with non-IID data partitions and standard model architectures (TCN for HAR, ResNet56 for CIFAR-10).

**Setup for Human Activity Recognition**    We utilized the UCI Human Activity Recognition (HAR) Dataset (Reyes-Ortiz et al., 2013), a widely used dataset in activity recognition research, especially in FL (Concone et al., 2022; Tu et al., 2021). The dataset includes sensor data from 30 subjects (aged 19–48) performing six activities: walking, walking upstairs, walking downstairs, sitting, standing, and laying. The data was collected using a Samsung Galaxy S II smartphone worn on the waist, capturing readings from both the accelerometer and gyroscope sensors. Each subject in the dataset was treated as an individual client in the FL setup, preserving the data's unique activity patterns and non-IID nature. We allocated 70% of each client's data for training using 5-fold cross-validation and 30% for testing, enabling evaluation of the model on independently collected test data. Data preprocessing involved applying noise filters to the raw signals and segmenting the data using a sliding window approach with a window length of 2.56 seconds and a 50% overlap, resulting in 128 readings per window. We selected the HAR dataset for evaluation F*in*P due to its inherited non-IID structure.

We trained the model in a federated learning setting using the Federated Averaging Algorithm (FedAvg) aggregation method over 20 global communication rounds. Each client trained locally with a batch size of 64, a learning rate of 0.001 using Adam optimizer, 1 local epochs per round, and an impact factor $\beta$ of 2. These parameters ensured balanced model updates from each client while maintaining computational efficiency across the federated network. Each local model (one per subject) analyzes its time-series sensor data using Temporal Convolutional Network (TCN) modelBai et al. (2018). The TCN model, designed for time-series data, uses causal convolutions to capture temporal dependencies while preserving sequence order. The architecture includes two convolutional layers, each followed by max-pooling, with a final fully connected layer for classifying the six activity classes.

**Setup for CIFAR-10**    The CIFAR-10 dataset consists of 60000 32x32 color images in 10 classes, with 6000 images per class. There are 50000 training images and 10000 test images. We use the Dirichlet distribution $Dir(\alpha)$ to divide the CIFAR-10 dataset into $K$ unbalanced subsets similar to previous work in the literature (Mendieta et al., 2022; Hu et al., 2021), with $\alpha = 0.5$. Figure 2 demonstrates how the data are distributed among clients with $\alpha = 0.1$. We created 10 clients and employed ResNet56 (He et al., 2016) as the local model. Similar to the setup in HAR, we trained

the model over 20 global communication rounds. Each client is trained locally with an impact factor $\beta$ of 0.1 as described in Equation 2. As for ablation experiment in $\beta$, we evaluated multiple values of $\beta = \{0.05, 0.1, 0.3, 0.5\}$ described in Equation 2. We used the same CNN proposed in Hu et al. (2023) as the local model.

**Experiment setup**    All experiments were conducted using a single NVIDIA A30 GPU with 24 GB memory.

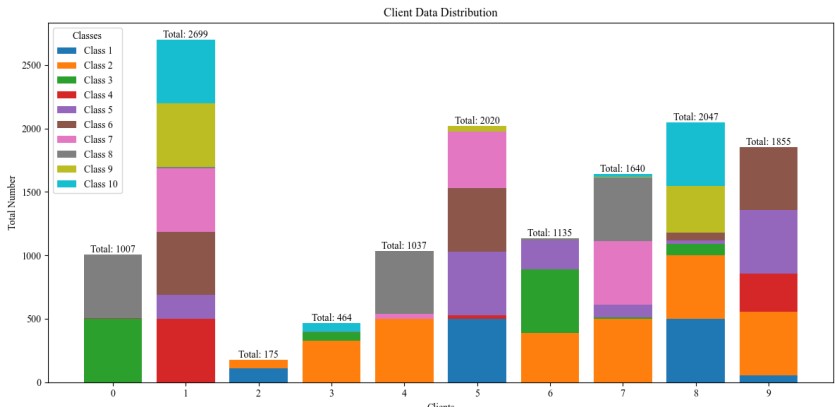

Figure 2: CIFAR dataset profile for each client after Dirichlet sampling with $\alpha = 0.1$

# C    EXPERIMENT RESULTS ON HAR

## C.1    DISPARITY OF SIA ACCURACY AND PREDICTION LOSS ON TARGET RECORDS AMONG CLIENTS

Figure 3 and Figure 4 show the disparity of SIA accuracy and prediction loss across clients using HAR dataset during 20 rounds of FL training with PCA aggregation.

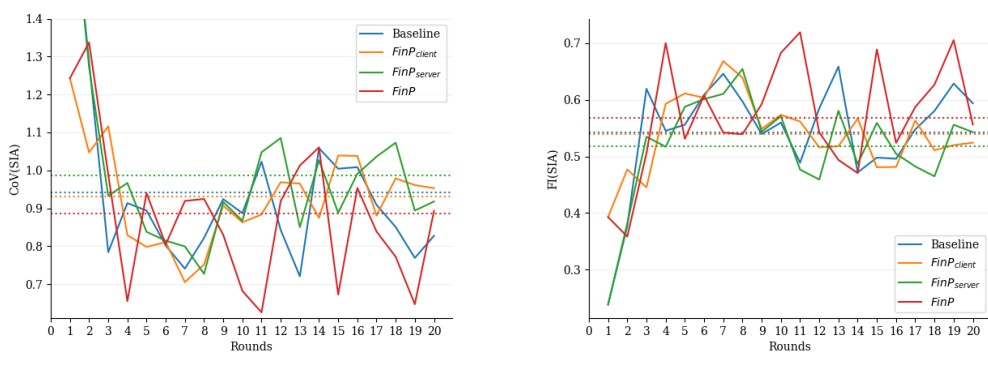

(a) Coefficient of variation for SIA accuracy CoV(SIA$_{acc}$).    (b) Fairness index of SIA accuracy FI(SIA$_{acc}$).

Figure 3: Disparity of SIA accuracy among clients using HAR dataset.

## C.2    CLASSIFICATION ACCURACY

Figure 5 demonstrates the classification accuracy of Baseline, F$in$P$_{client}$, F$in$P$_{server}$, and F$in$P across 20 rounds of training in FL showing the convergence round for each approach with PCA aggregation.

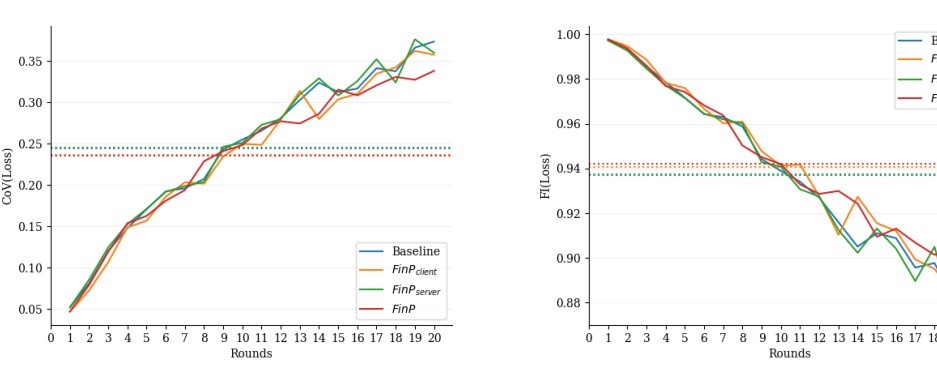

(a) Coefficient of variation for the prediction loss CoV(Loss).  (b) Fairness index of the prediction loss FI(Loss).

Figure 4: Disparity of prediction loss among clients using HAR dataset.

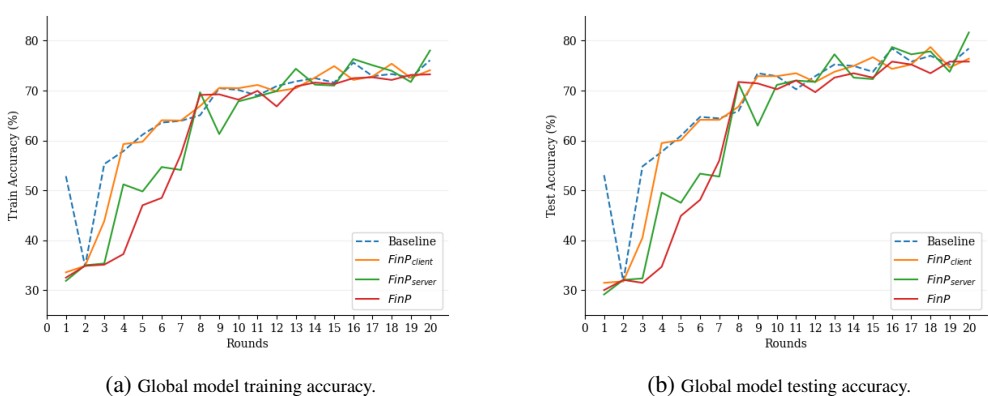

(a) Global model training accuracy.  (b) Global model testing accuracy.

Figure 5: Global model classification accuracy using HAR dataset.

### C.3 COMPUTING TIME EFFICIENCY

As for computation time efficiency, F$in$P requires more time on training due to PCA calculation involved in aggregation on the server-side. In particular, the computation time per round in the federated learning setup of the Baseline and F$in$P with PCA aggregation is $7s$ and $116s$, respectively. PCA calculation takes majority of the total $116s$ in F$in$Pwith PCA. By using Adaptive lightweight aggregation in F$in$P, it takes $19s$, around $17\%$ of PCA aggregation needed. However, the training time may be reduced by utilizing numerical PCA acceleration methods (Halko et al., 2011; Ross et al., 2008; Williams & Seeger, 2001) Besides, F$in$P maintains comparable convergency, with only 1 round late compared to the Baseline.

### C.4 EFFECT SERVER-SIDE COMPONENT AND CLIENT-SIDE COMPONENT IN F$in$P

Using adaptive weighting for aggregation on the server-side reduced the disparity of the PCA distance across all clients as observed in Figure 6. This is accounted to the adaptive weights in each FL round to reduce this disparity. The different weights assigned to every client model parameters during aggregation in every FL round are shown in Figure 7a.

We evaluated the relation between the top Hessian eigenvalue ($\lambda_{max}$) and the Hessian trace ($H_T$) to use them to compute the relative overfitting ranking across clients. As seen inFigure 8 they are strongly bonded with Spearman's rank correlation coefficient $\approx 1$. Hence, those two values are used with the same weight to compute relative ranking as an adaptive controlling regularization strength $\rho$ for each client across 20 rounds in FL. Figure 7b shows how the value of $\rho$ changes over the 20 rounds for each client.

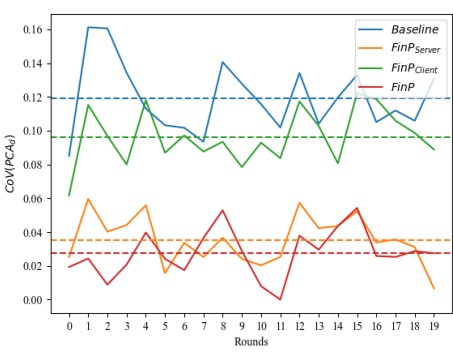 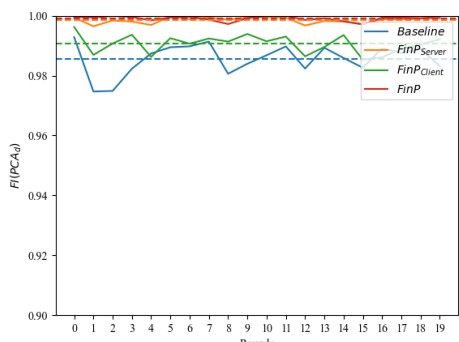

(a) Coefficient of Variation of the PCA distance to the global model CoV(PCA$_d$).

(b) Fairness Index of PCA to the global model FI(PCA$_d$).

Figure 6: Disparity of PCA distance between the global model and the client models using HAR dataset.

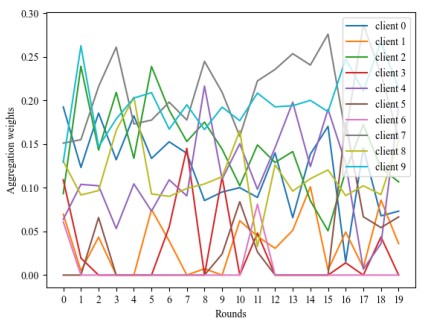 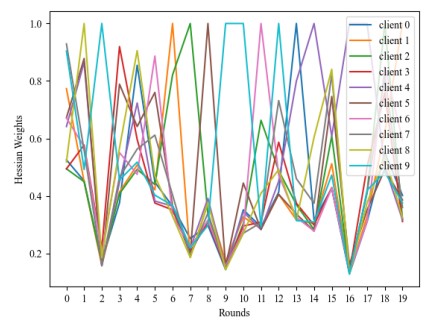

(a) Aggregation weights $\mathcal{W}$ in Federated learning.

(b) Relative overfitting rank $\rho_k$.

Figure 7: Aggregation weights $\mathcal{W}$ in Federated learning and adaptive controlling regularization strength based on the relative overfitting rank $\rho_k$ in the HAR dataset experiment.

# D    EXPERIMENT RESULTS ON CIFAR-10

## D.1    DISPARITY OF SIA ACCURACY AND PREDICTION LOSS ON TARGET RECORDS AMONG CLIENTS

Figure 11 and Figure 16b show the disparity of SIA accuracy and prediction loss across clients using CIFAR-10 dataset during 20 rounds of FL training with PCA aggregation using ResNet as the client local model.

## D.2    CLASSIFICATION ACCURACY

Figure 13 demonstrates the classification accuracy of Baseline, FedAlign, and F*in*P across 20 rounds of training in FL with PCA aggregation showing the convergence round for each approach.

## D.3    ABLATION ON IMPACT FACTOR $\beta$ IN CIFAR-10 WITH CNN

We conduct an ablation study on the hyperparameter $\beta$. The results are summarized in Table 3. The impact factor $\beta$ regularized the impact of the Lipschitz constant, where a larger $\beta$ improves the fairness regularization on client training loss. We evaluated experiments with $\beta$=0.05, 0.1, 0.3, and 0.5. Compared with the Baseline Hu et al. (2023), we found that increasing $\beta$ can achieve better fairness on SIA confidence by improving the FI(Loss) from 0.83 to 0.87 with $\beta$=0.3. Furthermore, F*in*P can reduce the average Mean(SIA$_{acc}$), and Max(SIA$_{acc}$) from 40.91%, and 46.7% to 31.85%, and 39.90% respectively. A larger $\beta$ can also improve testing accuracy due to improved generalization. However, excessive $\beta$ can be harmful to the model convergence as the Lipschitz constant

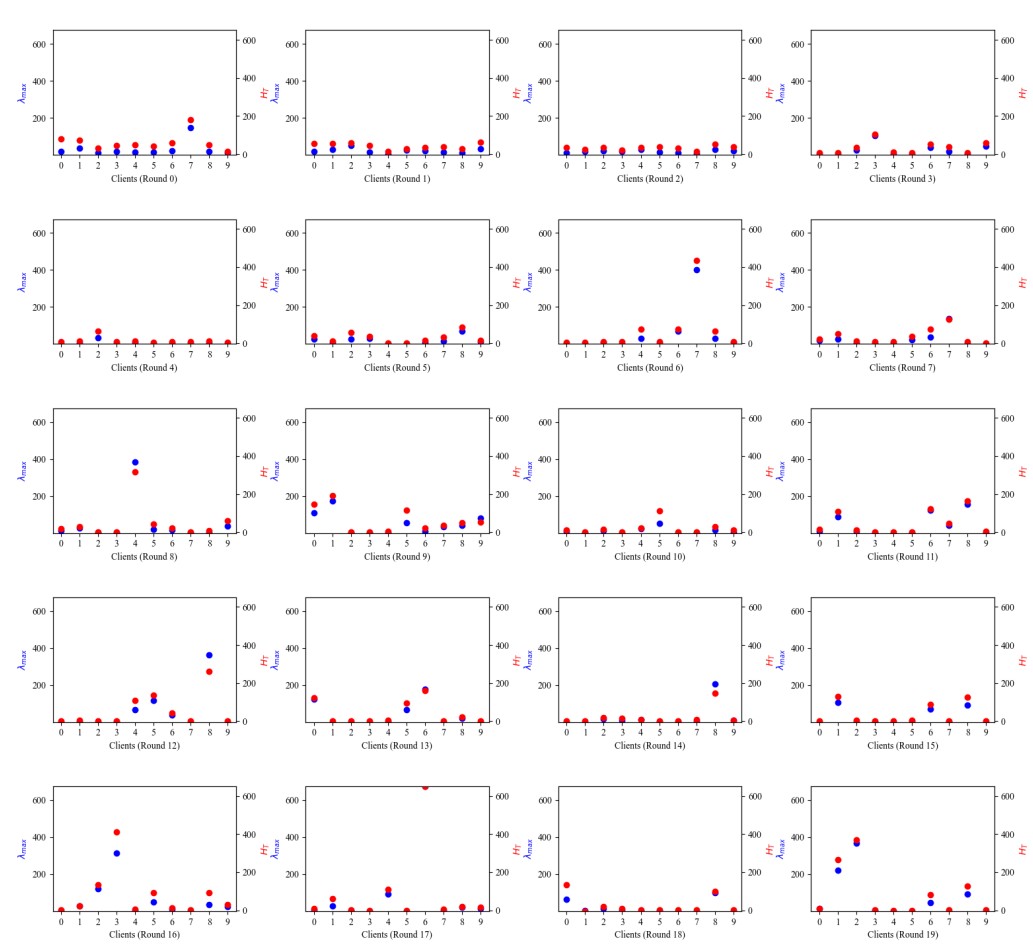

Figure 8: Scatter figures for top Hessian eigenvalue ($\lambda_{\max}$) and Hessian trace ($H_T$) across rounds using HAR dataset. They are strongly bonded.

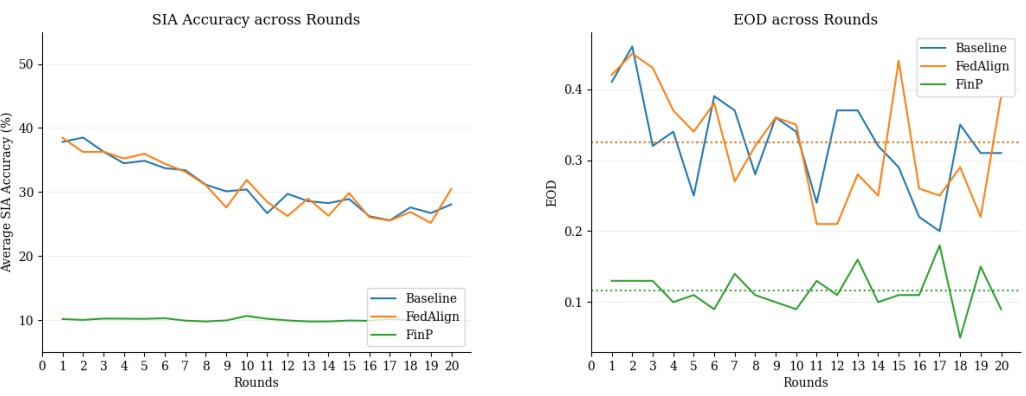

Figure 9: Average SIA accuracy in CIFAR-10.

Figure 10: EOD in CIDAR-10.

may dominate the client training loss as demonstrated in an experiment with $\beta$=0.5. Thus, a proper impact factor is essential to improve fairness without compromising convergence.

The effect of $\beta$ on training and testing accuracy during the 20 rounds in FL are shown Figure 14. At $\beta = 0.5$ the global model could not converge to acceptable accuracy. More visual illustrations for the effect of $\beta$ using CNN as a local client model are shown in Figures 15, 16, 17 and 18. We can

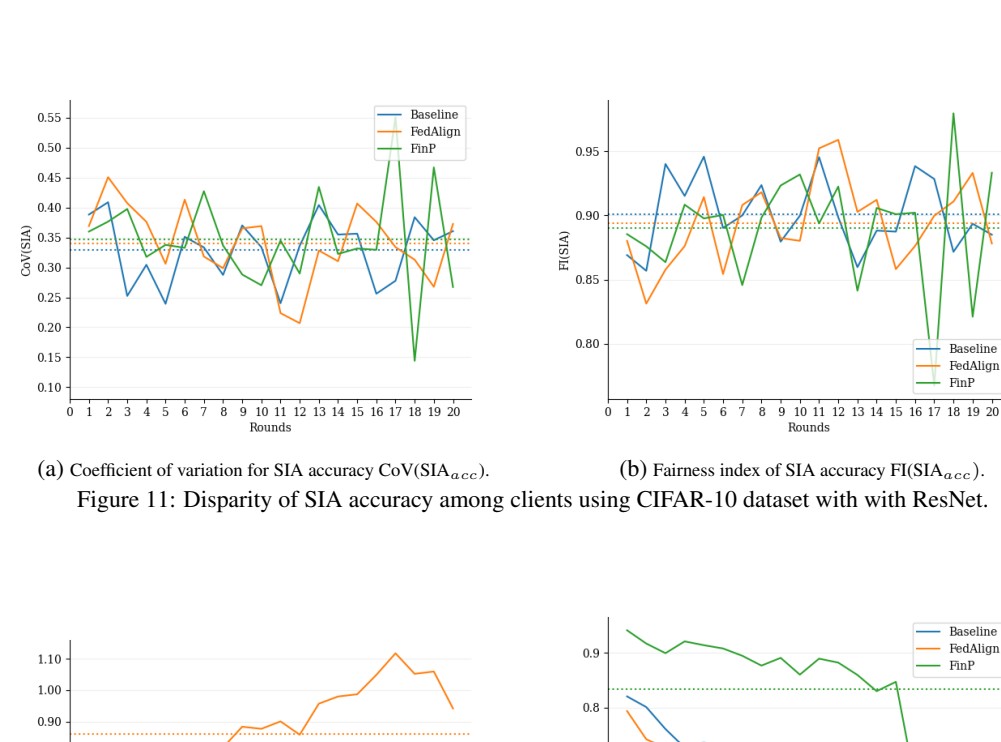

(a) Coefficient of variation for SIA accuracy CoV(SIA$_{acc}$).

(b) Fairness index of SIA accuracy FI(SIA$_{acc}$).

Figure 11: Disparity of SIA accuracy among clients using CIFAR-10 dataset with with ResNet.

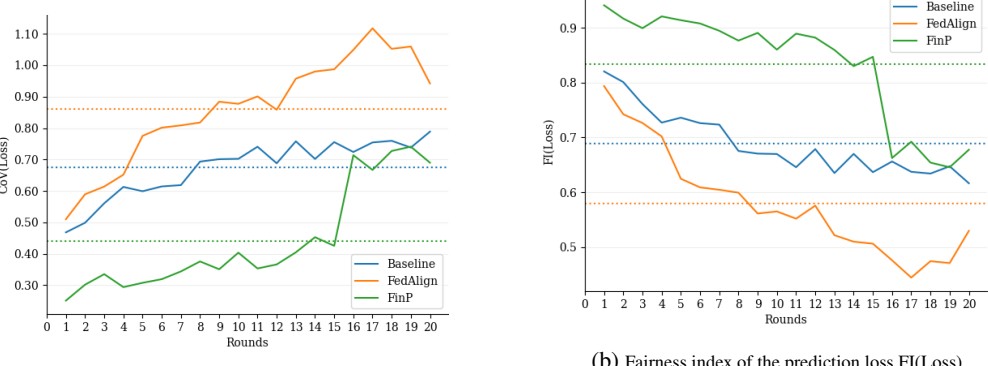

(a) Coefficient of variation for the prediction loss CoV(Loss).

(b) Fairness index of the prediction loss FI(Loss).

Figure 12: Disparity of prediction loss among clients using CIFAR-10 dataset with ResNet.

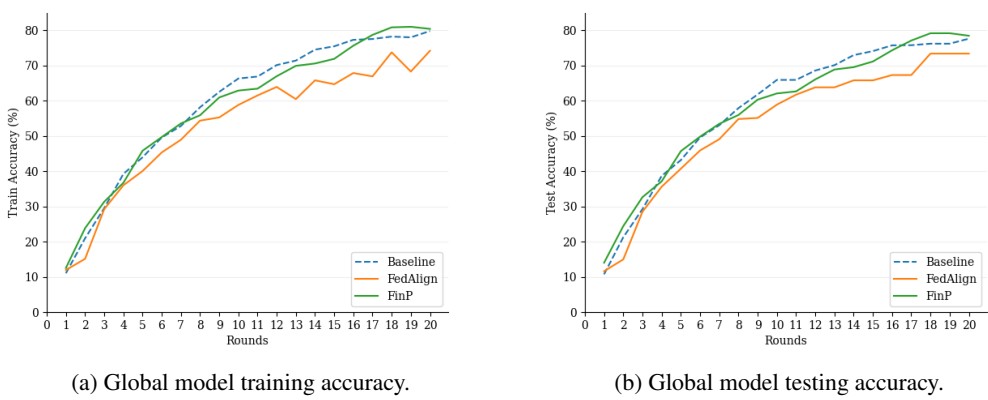

(a) Global model training accuracy.

(b) Global model testing accuracy.

Figure 13: Global model classification accuracy using CIFAR-10 with ResNet.

observe that as the model could not converge at $\beta = 0.5$, it causes the lowest average SIA accuracy as seen in Figure 17. However, this does not indicate better fairness as shown in Figure 18.

The computing time efficiency will not change much with different values of $\beta$ as the main factor of computation time in our implementation comes from computing PCA as explained in Appendix C.3.

Table 4: Ablation experiments on $\beta$. Results of CIFAR dataset using CNN as the local model Hu et al. (2023).

| | Accuracy (%) | | Privacy Metrics (%)) | | Fairness Metrics | | | Efficiency |
|---|---|---|---|---|---|---|---|---|
| | Training | Testing | Mean(SIA$_{acc}$)↓ | Max(SIA$_{acc}$)↓ | CoV(SIA$_{acc}$)/FI(SIA$_{acc}$) | CoV(loss)/FI(Loss) | EOD↓ | Converge |
| Baseline Hu et al. (2023) | 75.62 | 62.37 | 40.91 | 46.70 | 0.23/0.95 | 0.46/0.83 | 0.29 | round 5 |
| F*in*P($\beta$=0.05, PCA) | 69.31 | 59.67 | 39.51 | 43.40 | 0.21/0.96 | 0.46/0.83 | 0.27 | round 5 |
| F*in*P($\beta$=0.1, PCA) | 70.45 | 61.19 | 39.47 | 43.70 | **0.19/0.96** | 0.44/0.84 | **0.25** | round 7 |
| F*in*P($\beta$=0.3, PCA) | 71.17 | 63.81 | **31.85** | **39.90** | 0.31/0.90 | **0.38/0.87** | 0.31 | round 12 |
| F*in*P($\beta$=0.5, PCA) | 10 | 10 | N/A | N/A | N/A | N/A | N/A | N/A |
| F*in*P($\beta$=0.05, ALA) | **76.03** | **64.26** | 38.62 | 43.90 | 0.23/0.95 | 0.46/0.83 | 0.29 | round 6 |
| F*in*P($\beta$=0.1, ALA) | 74.64 | 63.94 | 37.39 | 42.50 | 0.25/0.94 | 0.44/0.84 | 0.32 | round 7 |
| F*in*P($\beta$=0.3, ALA) | 71.00 | 64.09 | 34.09 | 41.00 | 0.34/0.89 | 0.41/0.86 | 0.38 | round 11 |

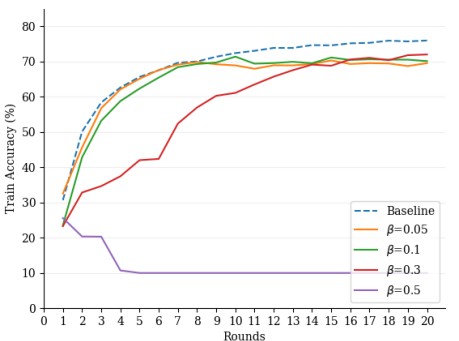

(a) Global model training accuracy.

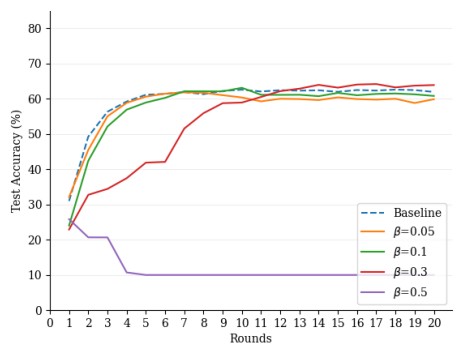

(b) Global model testing accuracy.

Figure 14: Ablation experiment of Global model classification accuracy using CIFAR-10 with CNN.

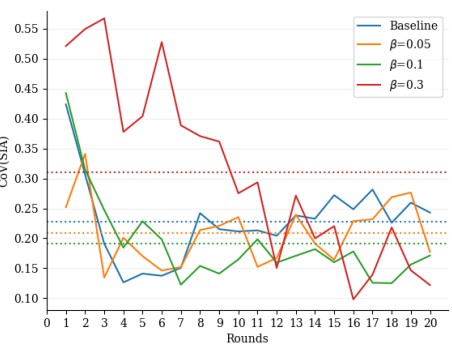

(a) Coefficient of variation for SIA accuracy CoV(SIA$_{acc}$).

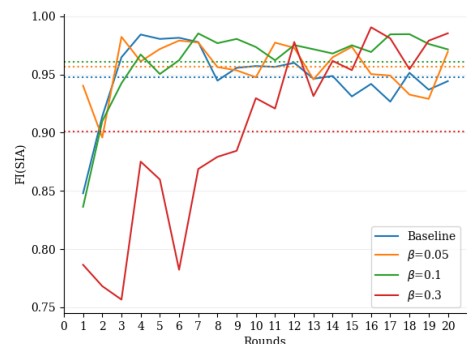

(b) Fairness index of SIA accuracy FI(SIA$_{acc}$).

Figure 15: Disparity of SIA accuracy among clients using CIFAR-10 dataset with base model CNN.

## D.4 COMPARISON ON SERVER SIDE PCA AND ADAPTIVE LIGHTWEIGHT AGGREGATION IN CIFAR-10 WITH CNN

Following the previous ablation experiments on the impact factor $\beta$, we changed the server side PCA aggregation to adaptive lightweight aggregation. The results are shown in Table 4 Adaptive lightweight aggregation uses overfitting level from client as the indicator to weighted-sum local models as global models. This method needs extra overfitting level information from clients unlike PCA aggregation but it avoids complex PCA calculation and impractical computation time on real world scenario. We observed the lightweight aggregation maintained the model convergency within 20 rounds in Figure 19. The improvement of FI(Loss)(Figure 21) and average SIA attack accuracy reduction(Figure 22) is comparable to utilizing PCA aggregation.

As we discussed in Section 4.2, adaptive lightweight aggregation method will not add any overhead in server side since there is no PCA calculation but need extra feedback on overfitting level $\rho_k$ from

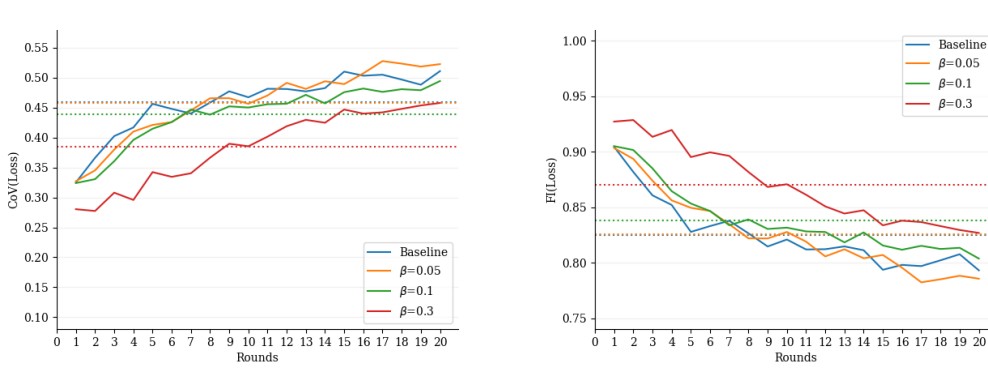

(a) Coefficient of variation for the prediction loss CoV(Loss).    (b) Fairness index of the prediction loss FI(Loss).

Figure 16: Disparity of prediction loss among clients using CIFAR-10 dataset with base model CNN.

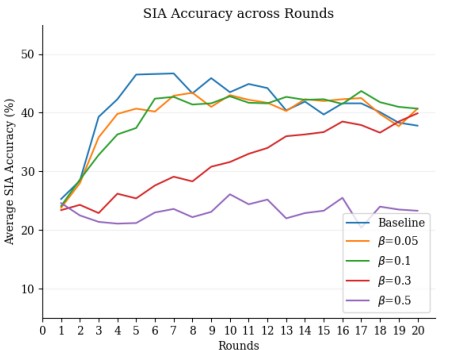
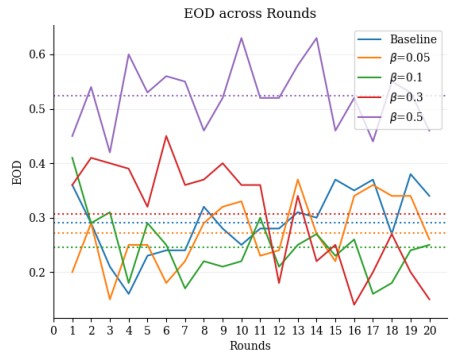

Figure 17: Average SIA accuracy of Baseline and different $\beta$ using CIFAR-10 dataset with CNN.

Figure 18: Equal opportunity difference (EOD) using CIFAR-10 dataset with CNN.

clients. The overfitting levels are calculated in client side and collected in server. PCA aggregation doesn't require any more information on overfitting level but takes a lot of time.

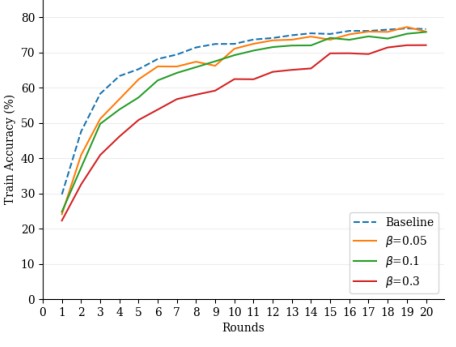
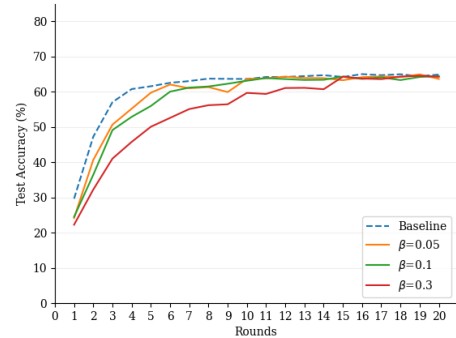

(a) Global model training accuracy.    (b) Global model testing accuracy.

Figure 19: Adaptive lightweight aggregation of global model classification accuracy using CIFAR-10 with CNN.

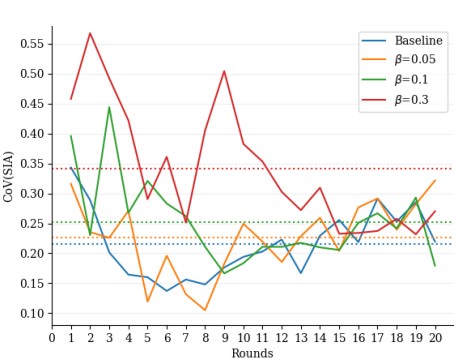
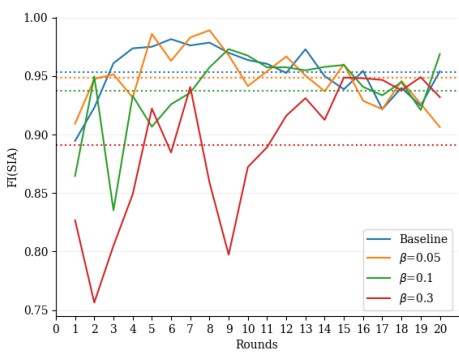

(a) Coefficient of variation for SIA accuracy CoV(SIA$_{acc}$).

(b) Fairness index of SIA accuracy FI(SIA$_{acc}$).

Figure 20: Adaptive lightweight aggregation disparity of SIA accuracy among clients using CIFAR-10 dataset with base model CNN.

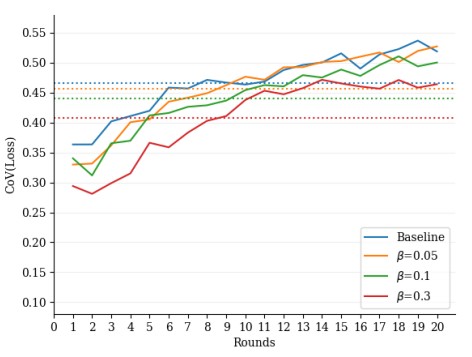
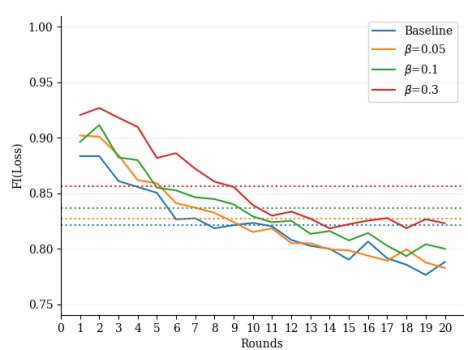

(a) Coefficient of variation for the prediction loss CoV(Loss).

(b) Fairness index of the prediction loss FI(Loss).

Figure 21: Adaptive lightweight aggregation disparity of prediction loss among clients using CIFAR-10 dataset with base model CNN.

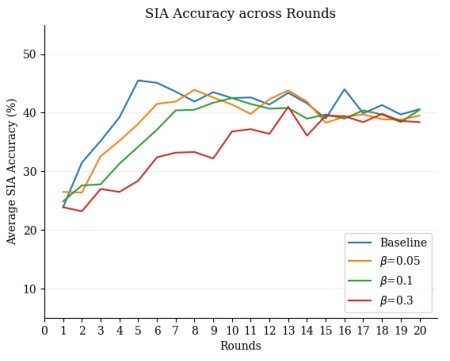
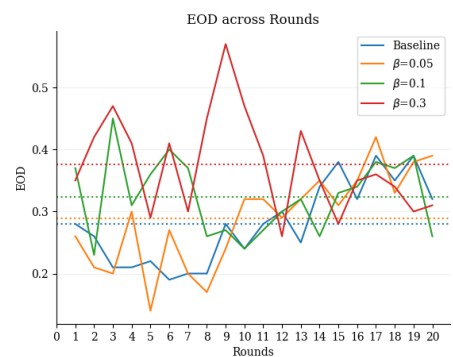

Figure 22: Adaptive lightweight aggregation average SIA accuracy of Baseline and different $\beta$ using CIFAR-10 dataset with CNN.

Figure 23: Adaptive lightweight aggregation equal opportunity difference (EOD) using CIFAR-10 dataset with CNN.

## D.5 ABLATION ON LOCAL TRAINING EPOCH

As more local training epochs can cause overfitting on local data per client, we evaluated the effect of the local training epoch on the average SIA accuracy for the baseline CNN on CIFAR-10. As shown in Figure 24, Our results show that Mean(SIA$_{acc}$) increases from 24.31%, to 40.91%, to 42.9% with local epochs 1, 5, 10, respectively. Similarly, Max(SIA$_{acc}$) increases from 27.20%, to

46.7%, to 51.4% with local epochs 1, 5, 10, respectively. As the local epoch (lp) increases, the overfitting increases impacting by the average SIA accuracy.

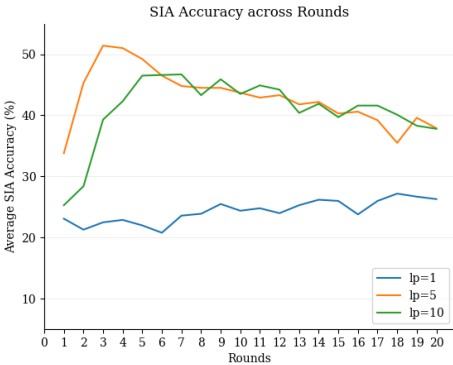

Figure 24: Average SIA accuracy on CIFAR-10 with CNN with varying number of local training epochs.