# OpenReview forum: "FinP: Fairness-in-Privacy in Federated Learning by Addressing Disparities in Privacy Risk"
_ICLR.cc/2026/Conference — ICLR 2026 Conference Withdrawn Submission_

### Official Review · Reviewer_XBYS · 2025-10-23

**Soundness:** 1
**Presentation:** 2
**Contribution:** 1
**Rating:** 2
**Confidence:** 4

**Summary:**

The paper looks at source inference attacks (SIA) in federated learning (FL), and proposed a method for essentially reducing the variability in the attack success between FL clients. The proposed approach adds local regularization (using an overfitting rank coefficient based either on the top Hessian eigenvalue and Hessian trace or a more compute efficient approximation, and on the spectral norm of the Jacobian) on client-side to avoid overfitting, as well as a server-side weighting based on the client overfitting coefficient (calculated either using PCA distance between model update and global model, or using the same ranking used for local regularization). The proposed method is compared to vanilla FedAvg and FedAlign, an existing method for FL in heterogeneous data settings, on Human activity recognition and CIFAR10 data sets using temporal CN, ResNet, and CNN models.

**Strengths:**

* The paper is mostly well-written and easy to read.
* To my knowledge, the proposed method is novel.

**Weaknesses:**

1) It is currently very hard to evaluate the significance of the empirical experiments (see Questions for details).

2) The proposed method requires significantly more communications/computations probably making it impractical for many real use cases.

3) There is no code available.

4) The paper is missing baselines based on differential privacy (DP). I find this unacceptable, as DP is currently the standard method for preventing privacy breaches such as MIAs, and has been extensively studied in FL as well (see Questions for some more comments on this). The related work discussion on DP (and to some extent fairness) seems pretty shallow.

5) Using only 2 datasets does not give too much evidence even in the best case. As the support for the method is mostly empirical, using more datasets (and models) would be more convincing.

6) The general claim that privacy issues are mostly due to overfitting. which is the basic motivation behind the method, seems odd. There is plenty of research showing that, eg, MIA performance is not only (or, depending on the case, even significantly) due to overfitting (see, eg, Dodd et al. 2025: The tail tells all for a very recent overview on the existing papers, also cf. Feldman et al. 2021: Does learning require memorization?).

**Questions:**

## Questions and further comments, in decreasing order of importance

1) Please clarify how the hyperparameters for the proposed method as well as for all the baselines have been tuned.

2) Please add, eg, standard error of the mean or other variability measure to all the results to convey some idea of how reliable the results are; eg, looking at Figs 5 & 13 in the Appendix, it looks like the (many of the) claimed results might be just due to randomness.

3) The statements and proofs of Prop 1,2 & 3 are very hand-wavy and vague, please formalize these properly to clarify what exactly you claim and when these claims hold.

4) If the privacy problem really is mostly due to local overfitting, it would make sense to try addressing it using some standard regularization technique, say, $\ell_2$, or add dropout layers, as these are computationally cheap, do not add any communications and should mitigate the issue. Have you run such experiments?

5) On the required amount of communications & compute, while there are some mentions on the required extra compute, please include the amount of extra communications and make this information more systematic and easier to find (eg, include $\mathcal O$ results for compute & amount of communications in C.3 separately for each of the possible components).

6) Do you plan to release the source code?

7) On DP (lines 463-470): the discussion on DP seems very much like a straw man (basically copy-pasted from Hu et al. 2023, who need to do some pretty weird things in the experiments to get basically random chance level accuracy with MNIST using a record-level $\varepsilon > 1, \delta=?$ approximate DP) just to avoid doing any experiments. So yes, taking a hit on utility is unavoidable if you want to provide meaningful privacy (so this is not just a limitation in DP, see, eg., Dinur and Nissim 2003 and [this blog](https://differentialprivacy.org/reconstruction-theory/) spelling it out). However, for something like CIFAR10 having meaningful protection with good utility is mostly achievable (see eg De et al. 2022: Unlocking high accuracy for a very good discussion & code in the centralized setting that should help setting a proper baseline); while data split details & DP assumptions could lead to unacceptably low utility, the results depend a lot on the details, which is why I think that running properly done experiments is crucial. As a side note, client-level DP is always at least as strong protection as record-level DP (so generally also has lower utility for any given privacy budget

---

### Official Review · Reviewer_scWW · 2025-11-01

**Soundness:** 2
**Presentation:** 3
**Contribution:** 3
**Rating:** 4
**Confidence:** 4

**Summary:**

This paper introduces FinP, a framework for fairness-in-privacy in federated learning (FL). Unlike prior work that primarily focuses on average privacy protection, FinP aims to equalize privacy risks across clients by mitigating disparities in vulnerability to source inference attacks (SIA). The method combines (1) server-side adaptive aggregation, which adjusts client contributions based on estimated privacy risk, and (2) client-side regularization, which penalizes overfitting using a Lipschitz-based term scaled by each client’s relative overfitting rank. Experiments on HAR and CIFAR-10 show that FinP can reduce disparities in SIA success rates and improve fairness metrics (e.g., Equal Opportunity Difference) with comparable accuracy.

**Strengths:**

- **Novel direction and concept - fairness in privacy.**  The paper opens a new and underexplored dimension in federated learning by introducing fairness considerations in privacy, beyond classical utility or accuracy fairness. If this is indeed the first formal treatment of “privacy fairness,” it is a substantial conceptual contribution.
- Idea of two-sided design where clients reduce the cause (overfitting) and server mitigates aggregation-level symptoms is interesting.
- The mathematical definitions of relative overfitting rank $\rho_k$ and its role in adaptive regularization are well presented. Theoretical propositions in Appendix A support the intuition that reducing Lipschitz constants can mitigate privacy vulnerability.

**Weaknesses:**

- **W1. Insufficient motivation and justification of introducing of "fairness-in-privacy".** While the idea is intriguing, the paper does not clearly articulate why fairness in privacy is necessary beyond minimizing worst-case leakage. The authors should elaborate on the ethical and practical implications of unequal privacy exposure (e.g., why variance in leakage matters rather than the worst case).
- **W2. Limited comparison with existing defenses mechanism againt SIA**. Experiments only compare against FedAvg and FedAlign, which are outdated. The paper should include comparisons with state-of-the-art SIA/MIA defense mechanisms
- **W3.** The calculation and exchange of $\rho_k$ and Hessian-based metrics introduce nontrivial complexity, but the paper does not quantify these costs. The runtime comparison (PCA: 116s vs. ALA: 19s per round) is helpful but insufficient to evaluate scalability in large-scale FL.
- **W4.** The observed gain is significant only for CIFAR-10 with ResNet, while improvements on HAR and smaller CNNs are modest. This suggests FinP may not generalize well across diverse data or architectures. The authors should discuss why the method benefits high-dimensional models disproportionately.
- **W5.** The PCA-based aggregation (used as the main FinPserver) can be expensive and unstable in high-dimensional neural networks. Although ALA is proposed as a lightweight relaxation, its fairness gains are notably smaller and lack theoretical backing.

**Questions:**

- Q1. What is the theoretical or practical justification for choosing variance-based fairness rather than minimizing the worst-case SIA success probability?
- Q2. How exactly is the fairness of privacy quantified beyond the reduction in CoV of SIA accuracy? Does FinP ensure that average privacy is maintained while improving fairness, or does it sacrifice stronger clients’ privacy to equalize risks?
- Q3. What is asymptotic computational and communication overhead of FinP (w.r.t important system parameters including the number of clients, model size, ...)?
- Q4. Could FinP be combined with existing privacy mechanisms (e.g., differential privacy or secure aggregation), or are these approaches incompatible due to conflicting noise or regularization effects?
- Q5. Why do fairness improvements appear substantially larger for CIFAR-10 + ResNet compared to HAR? Is this related to model expressiveness, dataset heterogeneity, or PCA stability?

---

### Official Review · Reviewer_WpUb · 2025-11-01

**Soundness:** 2
**Presentation:** 2
**Contribution:** 2
**Rating:** 2
**Confidence:** 3

**Summary:**

This paper presents FinP, a framework designed to improve fairness in privacy for Federated Learning (FL) systems. It argues that privacy risks are not evenly distributed among clients and aims to solve this by targeting disproportionate vulnerability to Source Inference Attacks (SIA). FinP uses a dual strategy: a server-side method to adaptively aggregate client models and a client-side regularization technique to help clients reduce their own vulnerability. The authors evaluate FinP on the HAR and CIFAR-10 datasets, showing it can improve fairness in privacy risk with a small impact on model accuracy.

**Strengths:**

1.  **Well-Motivated Problem:** The paper addresses the important topic of fairness in privacy. Moving beyond average privacy guarantees to consider the equitable distribution of risk is a valuable and necessary step for the responsible deployment of FL systems.

2.  **Logical Two-Part Design:** The proposed solution, which addresses the problem from both the server and client perspectives, is well-structured. This approach of correcting existing risk disparities at the server while also reducing their underlying causes at the client is a sensible design choice.

3.  **Detailed Experiments:** The evaluation is conducted across two datasets with different characteristics. The paper includes multiple relevant metrics for fairness and privacy, comparisons to other methods, and ablation studies that provide a good overview of the method's performance.

**Weaknesses:**

The paper presents an interesting approach to an important problem. However, there are some aspects related to the framework's practical application and underlying assumptions that could be strengthened.

1.  **Concerns Regarding Computational Cost and Scalability:** The primary server-side aggregation method is based on PCA distance. The experiments in the appendix (C.3) show that this method increases the per-round computation time from 7 seconds (baseline) to 116 seconds. This level of computational overhead could be a barrier to using the framework in real-world FL scenarios, particularly those involving many clients or frequent communication rounds. The proposed lightweight alternative (ALA) is faster but still adds noticeable overhead compared to the baseline.

2.  **Assumptions about the Federated Learning Environment:** The FinP framework's design relies on certain conditions that may be challenging to meet in all FL settings.
    *   **Client-Side Requirements:** The framework requires clients to modify their local training procedure to include a specific Lipschitz-based regularization. This adds complexity on the client side and assumes that all clients are able and willing to implement this change.
    *   **Server-Side Calculations:** To provide feedback to clients, the server must compute Hessian properties for each client's model. This can be a demanding task, adding to the server's computational load each round.
    *   **Communication Loop:** The framework depends on a consistent feedback loop where the server sends information to clients, and clients use it to adjust their training. This assumes a relatively stable and synchronous environment.

3.  **Reliability of the PCA Distance Metric:** As the authors note in Section 4.2, using PCA distance as a proxy for privacy risk can be "fragile and unreliable" for high-dimensional models. This raises questions about how consistently the method would perform with the complex neural networks often used in modern FL applications. The framework's effectiveness appears linked to the reliability of this proxy metric.

**Questions:**

1. Could you characterize the scalability limits of the PCA-based aggregation?
2. Given that PCA distance can be sensitive in high-dimensional spaces, how did you select the number of principal components for your analysis?

---

### Official Review · Reviewer_wF6j · 2025-11-04

**Soundness:** 2
**Presentation:** 3
**Contribution:** 3
**Rating:** 4
**Confidence:** 4

**Summary:**

The paper examines  fairness inherent in federated learning privacy risks, more particularly: source inference attacks (SIA) which can identify which client contributed a record. FinP addresses this two ways: (1) client-side Lipschitz/Jacobian regularization scaled by relative overfitting rank ρₖ computed from Hessian eigenvalues/trace, and (2) server-side adaptive aggregation minimizing variance of per-client privacy risk via PCA distance or lightweight ALA weighting. Evaluation on HAR (30 clients) and CIFAR-10 (10 clients, 20 rounds) is done on ResNets and CNN.

**Strengths:**

1) Clear fairness framing in terms of metrics: Operationalizes per-client SIA vulnerability as fairness via CoV, EOD, and Fairness Index with concrete metrics
2) Two-sided design + strong CIFAR results:  drives Mean SIA near random while improving EOD, fairness indices, and accuracy

**Weaknesses:**

1)My main concern remains in how the privacy problem is done. Overfitting as a central cause of enhanced privacy risk is not something that is widely known (the authors also cite one paper). The framing seems to assume that this is a known phenomenon and the authors algorithm builds on top of it.Perhaps the author can denote a section recalling this connection between overfitting and privacy phenomenon and explaining it a bit more.

More importantly, If generalization and privacy are correlated  then how does the work of authors stand out to methods that are targeting better generalization in heterogeneous distributions. Isn't privacy implicit in those works. The framing as it stands seems shallow to me.

2) The relative rank which is based on max Eigen value and sum of Eigen values(Trace) does give information about the curvature of the local loss so perhaps controlling it across clients may improve generalization (though we know this is not a causal phenomenon)  but stretching it to privacy seems a bit abrupt. The flow from the problem to solution seems all of a sudden to me. Also are there implicit DP bounds in the proposed algorithm?

3) It is fine that the Eigen values are communicated but this opens up another risk which is an honest but curious may inquire more information from these transferred Eigen values about the local client. How are the authors resolving it?

4) The entire optimization problem relies on extra rounds of communication right? -- this should be discussed in the paper ie: comparing to FedAvg.

5) The existing baseline is standard FedAvg. Are there no works that look at SIA in federated settings?

**Questions:**

Weaknesses and Questions merged

---

### Note · Authors · 2025-11-14

**Comment:**

We have decided to withdraw this submission. We believe the manuscript will benefit significantly from more work based on the feedback we received. We are grateful to the reviewers for their detailed comments, which have given us a clear path for improvement.

**Withdrawal Confirmation:**

I have read and agree with the venue's withdrawal policy on behalf of myself and my co-authors.